# Earliest Olduvai hominins exploited unstable environments ~ 2 million years ago

Julio Mercader [1,2✉], Pam Akuku[3,4], Nicole Boivin [1,2,5,6], Revocatus Bugumba[7], Pastory Bushozi[8], Alfredo Camacho[9], Tristan Carter [10], Siobhán Clarke [1], Arturo Cueva-Temprana [2], Paul Durkin[9], Julien Favreau [10], Kelvin Fella[8], Simon Haberle [11], Stephen Hubbard [1✉], Jamie Inwood[1], Makarius Itambu[8], Samson Koromo[12], Patrick Lee[13], Abdallah Mohammed[8], Aloyce Mwambwiga[1,14], Lucas Olesilau[12], Robert Patalano [2], Patrick Roberts [2,5], Susan Rule[11], Palmira Saladie[3,4], Gunnar Siljedal[1], María Soto [15,16✉], Jonathan Umbsaar[1] & Michael Petraglia [2,5,6]

Rapid environmental change is a catalyst for human evolution, driving dietary innovations, habitat diversification, and dispersal. However, there is a dearth of information to assess hominin adaptions to changing physiography during key evolutionary stages such as the early Pleistocene. Here we report a multiproxy dataset from Ewass Oldupa, in the Western Plio-Pleistocene rift basin of Olduvai Gorge (now Oldupai), Tanzania, to address this lacuna and offer an ecological perspective on human adaptability two million years ago. Oldupai's earliest hominins sequentially inhabited the floodplains of sinuous channels, then river-influenced contexts, which now comprises the oldest palaeolake setting documented regionally. Early Oldowan tools reveal a homogenous technology to utilise diverse, rapidly changing environments that ranged from fern meadows to woodland mosaics, naturally burned landscapes, to lakeside woodland/palm groves as well as hyper-xeric steppes. Hominins periodically used emerging landscapes and disturbance biomes multiple times over 235,000 years, thus predating by more than 180,000 years the earliest known hominins and Oldowan industries from the Eastern side of the basin.

[1] University of Calgary, Alberta, Canada. [2] Max Planck Institute for the Science of Human History, Jena, Germany. [3] Institut Català de Paleoecologia Humana i Evolució Social (IPHES), Tarragona, Spain. [4] Àrea de Prehistòria, Universitat Rovira i Virgili (URV), Tarragona, Spain. [5] School of Social Science, University of Queensland, Saint Lucia, QLD, Australia. [6] National Museum of Natural History, Smithsonian Institution, Washington, DC, USA. [7] Ministry of Natural Resources and Tourism, Dar es Salaam, Tanzania. [8] University of Dar es Salaam, Dar es Salaam, Tanzania. [9] University of Manitoba, Winnipeg, Manitoba, Canada. [10] McMaster University, Hamilton, Ontario, Canada. [11] Australian National University, Canberra, ACT, Australia. [12] University of Iringa, Iringa, Tanzania. [13] University of Toronto, Toronto, Ontario, Canada. [14] National Natural History Museum, Arusha, Tanzania. [15] Madrid Institute for Advanced Study, Madrid, Spain. [16] Universidad Autónoma de Madrid, Madrid, Spain. ✉email: mercader@shh.mpg.de; shubbard@ucalgary.ca; sotoquesadamaria@gmail.com

Hominins underwent major biological transitions in the early Pleistocene, along with an increased reliance on stone tool use[1], overall dietary diversification[2–6], and long-distance dispersal[7–9]. The behavioural context of these shifts remains elusive due to a dearth of high-resolution chronostratigraphic and environmental datasets in direct association with Oldowan remains, available only for a small number of sites[10–17]. Furthermore, extrapolating offsite palaeoecological proxies from penecontemporaeneous boreholes and lake-drilling sequences[18] has limited applicability for understanding localized land use, the synchronous/diachronous occupation of varied terrestrial environments, and targeted habitat exploitation by Oldowan hominins.

Palaeoenvironmental reconstruction from sites 2.6 to 1.9 Ma has relied on indirect approximations of past vegetation from fauna and/or stable isotopes[13–17,19,20], denoting variably open grassland and forest mosaics in fluvial settings often from restricted stratigraphic intervals[21,22]. Our research reconciles the earliest Oldowan stone tools from Oldupai Gorge, a key complex for the study of hominin lifeways[23,24], with multiproxy datasets in direct association with stratified archaeological and fossil assemblages, recording episodic exploitation of the same place in varied geomorphic contexts and sedimentary deposits. This dataset stands as a model of cross-disciplinary research to clarify the environmental context of early Oldowan sites. Here we examine hominin behavior in association with faunal and plant communities and provide evidence of vegetation physiognomy and cover from phytolith analysis and palynology, isotopic n-alkane values from plant waxes, stable isotopes from enamel, and regional fires from microcharcoal concentrations.

## Results

**Stratigraphy and archaeology.** The Ewass Oldupa site (Geological Locality 63, Fig. 1)[23] is located 350 m northeast of Geological Locality 64, where a dentally complete maxilla and lower face of *Homo habilis* (OH 65)[22] was recovered from strata dated to ~1.82 Ma[25,26]. At Ewass Oldupa, we exposed a thick sedimentary sequence with ages bracketed by existing $^{40}Ar/^{39}Ar$ dates of geochemically fingerprinted tuffs[26,27], which was further constrained at six localities along a 2 km transect (Supplementary Fig. 1, Table 1). The lowermost stratigraphic unit is the Naabi ignimbrite of the Ngorongoro Formation[18] (Figs. 1 and 2a). Above this sits the highly heterogeneous Coarse Feldspar Crystal Tuff compositional zone (CFCTcz)[28] (Figs. 1 and 2c), including the Coarse Feldspar Crystal Tuff (2.015 ± 0.006 Ma)[25]. Lower Bed I starts with the deposition of a green waxy clay[29] and intercalated carbonate beds, indicating lake expansion 1 km further west and occurring earlier than previously recorded[18,26–29] (Figs. 1–2e). Tuff IA (~2.0 Ma)[18] (Figs. 1 and 2f, g) contains characteristic Mg-rich augite, reworked basement-derived detrital grains and CFCTcz materials[25–28]. Upper Bed I includes Tuff IB (1.848 ± 0.003 Ma)[25]. Directly overlying this are waxy, green-brown claystones deposited during a period of high lake level (Figs. 1, 2h). Overlying these clays, a weakly stratified siltstone has the geochemical signature of Tuff IC (1.832 ± 0.003 / 1.848 ± 0.008 Ma)[22,25,26]. Capping it, a weakly planar stratified sandstone is geochemically consistent with Tuff ID, identified elsewhere in the western gorge as the interval containing the remains of *Homo habilis* (OH65)[22,26]. A thin bed with carbonate nodules and coarse-grained siliciclastic material is petrographically and geochemically identified as the Ng'eju Tuff (1.818 ± 0.006 Ma)[25,26].

The geo-archaeological record through the Ewass Oldupa sequence shows evidence of human occupation 2.0–1.9 Ma through rapidly shifting depositional environments that comprised meandering streams cutting across a volcaniclastic fan, followed by mass flow events, several transgression/regression lacustrine cycles, large and small prograding fluvial systems, with intercalated volcanism. Importantly, this record of hominin habitats predates, to our knowledge, the oldest fossiliferous and tool bearing deposits from the Eastern basin's locality of 'Douglas Korongo' (DK), which traditionally represented the earliest Oldowan presence in the region at ~1.848 ± 0.003 Ma[24,30,31].

Excavations at Ewass Oldupa recovered 1373 fossil specimens and 565 stone artefacts (Supplementary Table 1) showing consistent technological features from immediately post-Naabi to the base of Bed II. All knapping stages are present (Supplementary Table 1). Lithics derive from tabular, medium grained, grayish/whitish quartzite slabs that are clustered (7/m²) in discrete archaeo-stratigraphic assemblages (7/40 m²) (Figs. 3 and 4). Flakes are the dominant product (>60%), with spheroids and percussive materials being uncommon. There is no differential management of knapped surfaces, thus the technological systems do not follow hierarchical reduction strategies. Unidirectional and multidirectional knapping of cores was maintained within the constraints of natural shapes and angles for striking. Of the seven exploitation methods registered in 58 cores, multipolar-multifacial (number of cores = 19, median number of extractions = 9), unipolar-longitudinal (*n* = 15, median = 4), and orthogonal-bifacial (*n* = 13, median = 5) dominate (Supplementary Fig. 2c). Intensity of lithic reduction is inferred from the minimum number of extractions per core, which ranges from two to 16 removals. Overall, the simplicity of Ewass Oldupa's technical repertoire is shared with other early Oldowan assemblages[6,10–12,14,15] to include characteristics such as free-hand hard hammer percussion and multi-facial reduction. Principal Component Analysis (PCA) (Supplementary Fig. 2a, b, Supplementary Fig. 3) conveys that Ewass Oldupa's Oldowan shares many technical attributes with Kanjera South[14], Fejej[11], and to some extent Frida Leakey Korongo Zinj (FLK Zinj)[24]. Significantly, several aspects of Ewass Oldupa's stone industries (unifacial and bipolar cores, number of extractions, flake size, angular fragments, and scarcity of percussion tools) best match the techno-typological profile seen in older Oldowan sites (Gona 10 and Gona 12)[15]. In addition, PCA brings out a mosaic of early and classic Oldowan traits in Ewass Oldupa's lithics, which fall between the oldest Oldowan > 2 Ma and younger assemblages ≤1.85 Ma, where spheroids abound. Likewise, this PCA, inclusive of 18 assemblages and comprising 11 technical variables[15], demonstrates the outlier character of Lomekwi and its complete lack of affinity with Ewass Oldupa (Supplementary Figs. 2,3, Supplementary Table 2).

Comparison of geochemical fingerprinting of artefacts and regional rock outcrops (Supplementary Fig. 4) manifests hominins engaged in sourcing up to 12 km across the basin[32,33] while also exploiting quartzite locally, 400 m to the south (Naisiusiu). Macroscopically, four quartzite types of distinct colour, grain size, textural, and mineral composition were the preferred raw materials. Analysis of these varieties relative to reduction stage (Supplementary Fig. 2d) indicates that Oldowan groups targeted specific quartzite types for flakes, while other variants were for retouched tools. Spheroids and choppers were made of quartzite and ignimbrite, accessible from riverbeds.

**Early Oldowan ecology at ~ 2 Ma.** The earliest Oldowan presence at Ewass Oldupa, dating to 2.03 Ma, is in a post-eruptive landscape, occurring on an expansive ignimbrite flow[34] that buried the hydrological network of the southeast portion of the basin[18] while draping over metamorphic inselbergs in the west (Figs. 1a and 2a, b). Initial hominin use, documented in the form of 10 stone tools, appears 1 m above the Naabi ignimbrite and 17 m below Tuff IA. It occurred after landscape stabilisation in association with a

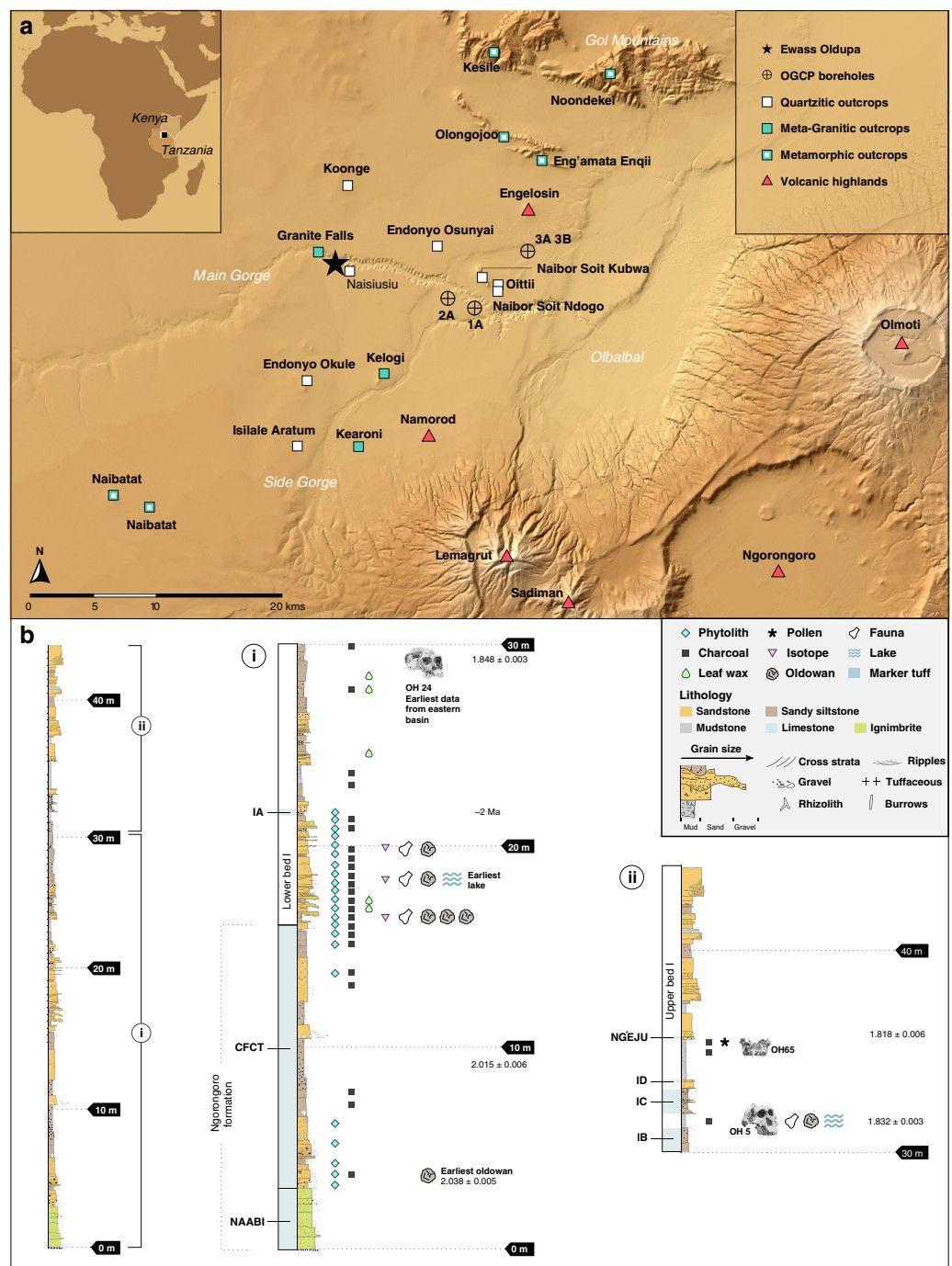

**Fig. 1 Geographic and Stratigraphic Context. a** Location map. Star symbol is for the site of Ewass Oldupa. Boreholes recently published[18] are shown for reference. Major rock types potentially available for hominin exploitation throughout the region are also marked. **b** Single stratigraphic section from Ewass Oldupa, subdivided into two segments: lower (i) and upper (ii). Ngorongoro formation: the succession starts with the Naabi, a green-gray, quartz-trachyte welded tuff dated to 2.038 ± 0.005 Ma[25]. A thin red diamictite overlies the Naabi and separates it from a series of fining-upward sandstone-dominated units composed of mafic tephra (Fig. 2b) that is capped by siltstone. Ngorongoro formation, CFCT compositional zone: the strata are composed of reddish-brown sandstone, volcanic detritus, and tephra/siliciclastic diamictite[28]. (i) Lower Bed I—six units that upward fine from pebble-cobble conglomerate to cross-stratified sandstone are attributed to meandering river channels, below Tuff IA. Subsequently, a fine-grained silty sandstone that fines into waxy, green-brown claystone with carbonate nodules was deposited in a lake setting. Lastly, interbedded sandstone, siltstone, claystone, and tuffaceous beds record fluctuating environments from lacustrine to shallow lakeshore to floodplain with small fluvial channels. (ii) Upper Bed I—starts with Tuff IB, overlying waxy, green-brown claystone (Fig. 2H), and then Tuff IC. Capping the succession is a beige, weakly stratified sandstone that is geochemically consistent with Tuff 1D[22,25,26]. Above this, the Ng'eju Tuff is encountered in a succession of fine-grained floodplain deposits. Additional stratigraphic information is presented in Fig. 5.

**Table 1 Average tuff mineral composition determined with electron microprobe analysis: the consistency between this study and previous studies supports the stratigraphic context of Ewass Oldupa.**

| Mineral | Tuff | Loc[b] | Sample | pop | n (meas.) | SiO$_2$ | TiO$_2$ | Al$_2$O$_3$ | FeO | MnO | MgO | CaO | Na$_2$O | K$_2$O | B$_2$O | SUM | Reference |
|---|---|---|---|---|---|---|---|---|---|---|---|---|---|---|---|---|---|
| Augite | IA | 65 | 02-T52f | pop 1 | 3 | 51.46 | 0.93 | 2.49 | 7.31 | 0.16 | 15.85 | 21.42 | 0.42 | n.d | n.d | 100.1 | 26 |
|  |  |  | St. Dev. |  |  | 0.63 | 0.29 | 0.24 | 1.92 | 0.06 | 1.02 | 0.88 | 0.13 | n.d | n.d | 0.22 |  |
|  | IA | 65 | 02-T52 | pop 2 | 3 | 54.21 | 0.33 | 0.34 | 3.79 | 0.1 | 18.22 | 21.88 | 0.39 | 0.02 | 0 | 100.29 | 26 |
|  | IAe | 61 | 12-L61-15 |  | 1 | 46.33 | 3.08 | 4.07 | 15.17 | 0.34 | 9.12 | 20.24 | 0.57 | 0.01 | n.d | 98.98 | 28 |
|  | IA | 63 | 63-1-6.8 |  | 12 | 50.52 | 1.25 | 3.32 | 8.43 | 0.21 | 14.48 | 21.22 | 0.69 | 0.01 | n.d | 100.5 | This study |
|  |  |  | St. Dev. |  |  | 1.96 | 0.66 | 1.55 | 2.4 | 0.14 | 1.86 | 1.19 | 0.71 | 0.01 | n.d | 0.25 |  |
|  | IB | 64 |  | pop 1 | 15 | 49.45 | 0.42 | 2.46 | 20.48 | 1.25 | 6.08 | 20.12 | 0.72 | n.d | n.d | 99.01 | 76 |
|  |  |  | St. Dev. |  |  | 2.14 | 0.03 |  | 0.83 | 0.16 | 0.54 | 0.53 | 0.17 |  |  | 2.04 |  |
|  | IB | 64 | 63-1-15.7 | pop 2 | 2 | 49.31 | 2.04 | 3.78 | 9.4 | 0.27 | 13.77 | 20.75 | 0.3 | 0.01 | n.d | 99.67 | 76 |
|  | IB | 63 | f | | 15 | 49.82 | 0.86 | 1.88 | 15.93 | 0.88 | 9.75 | 20.09 | 0.65 | 0.01 | n.d | 99.92 | This study |
|  |  |  | St. Dev. |  |  | 1.77 | 0.84 | 2.07 | 5.52 | 0.54 | 3.49 | 0.91 | 0.21 | n.d |  | 1.03 |  |
|  | IC | 64 | f | | 17 | 50.1 | 0.42 | 0.75 | 18.26 | 1.14 | 6.97 | 19.57 | 0.75 | n.d | n.d | 97.99 | 76 |
|  |  |  | St. Dev. |  |  | 0.58 | 0.09 | 0.21 | 2.79 | 0.08 | 2.02 | 0.43 | 0.15 | 0 | 0 | 0.54 |  |
|  | ICe | 66e | P8-19 | K1-K9 | 23 | 49.74 | 0.53 | 0.99 | 16.85 | 0.98 | 9.34 | 20.25 | 0.71 | 0 | 0.1 | 99.43 | 28 |
|  |  |  | St. Dev. |  |  | 0.78 | 0.35 | 0.55 | 3.65 | 0.33 | 2.6 | 0.35 | 0.26 | 0 | 0 | 1.07 |  |
|  | ICe | 62 | 14-L62-8 |  | 15 | 51.12 | 0.4 | 0.74 | 18 | 1.11 | 8.14 | 20.04 | 0.72 | 0 | 0 | 100.27 | 28 |
|  |  |  | St. Dev. |  |  | 0.5 | 0.12 | 0.19 | 2.29 | 0.12 | 1.68 | 0.45 | 0.1 | 0 | 0.01 | 0.63 |  |
|  | ICe | 61 | 14-L61-1.1 |  | 18 | 51.21 | 0.56 | 1.05 | 16.87 | 0.95 | 8.97 | 20.4 | 0.56 | 0.01 | 0 | 100.59 | 28 |
|  |  |  | St. Dev. |  |  | 0.35 | 0.32 | 0.63 | 2.8 | 0.29 | 2.03 | 0.5 | 0.06 | 0.01 | 0 | 0.49 |  |
|  | IC | 63 | 63-1-18.0 |  | 19 | 50.84 | 0.74 | 1.56 | 14.64 | 0.72 | 10.55 | 20.68 | 0.53 | 0.01 | n.d | 100.3 | This study |
|  |  |  | St. Dev. |  |  | 0.74 | 0.41 | 0.99 | 3.82 | 0.35 | 2.56 | 0.43 | 0.12 | 0.01 | n.d | 0.7 |  |
|  | ID | 64 |  |  | 16 | 52.62 | 0.61 | 1.21 | 14.61 | 0.75 | 9.94 | 20.58 | 0.45 | n.d | n.d | 100.8 | 76 |
|  |  |  | St. Dev. |  |  | 0.53 | 0.09 | 0.26 | 1 | 0.07 | 0.85 | 0.56 | 0.03 | 0 | n.d | 0.53 |  |
|  | IDe | 66e | P8-21 | K1-K5 | 23 | 48.96 | 0.56 | 1 | 14.93 | 0.79 | 10.43 | 21.07 | 0.53 | 0 | 0 | 98.32 | 28 |
|  |  |  | St. Dev. |  |  | 1.25 | 0.09 | 0.25 | 2.58 | 0.17 | 1.89 | 0.3 | 0.06 | 0 | n.d | 1.2 |  |
|  | ID | 64 | 64-3-9.9 |  | 11 | 50.51 | 0.61 | 1.39 | 14.86 | 0.79 | 10.46 | 20.61 | 0.55 | 0 | n.d | 99.82 | 26 |
|  |  |  | St. Dev. |  |  | 0.75 | 0.19 | 0.69 | 2.97 | 0.26 | 2.17 | 0.59 | 0.13 | 0 | n.d | 0.63 |  |
|  | Ng'eju | 64 | 02-T34 |  | 11 | 51.33 | 0.66 | 1.32 | 12.82 | 0.74 | 11.77 | 21.24 | 0.51 | 0 | n.d | 100.42 | 26 |
|  |  |  | St. Dev. |  |  | 0.53 | 0.19 | 0.32 | 0.94 | 0.12 | 0.76 | 0.23 | 0.07 |  |  | 0.49 |  |
|  | Ng'eju | 63 | 63-1-20.5 |  | 9 | 51.11 | 0.66 | 1.08 | 12.93 | 0.65 | 11.84 | 20.99 | 0.44 | 0 | n.d | 99.75 | This study |
|  |  |  | St. Dev. |  |  | 0.93 | 0.23 | 0.43 | 3.23 | 0.26 | 2.38 | 0.28 | 0.06 |  |  | 0.69 |  |
| Feldspar | IA | 65 | 02-T52f | pop 1 | 5 | 67.02 | n.d | 19.48 | 0.19 | n.d | n.d | 0.4 | 7.66 | 4.95 | 0.19 | 99.95 | 26 |
|  |  |  | St. Dev. |  |  | 1.09 |  | 0.47 | 0.16 |  |  | 0.38 | 0.43 | 0.66 | 0.16 | 0.99 |  |
|  | IA | 65 | 02-T52f | pop 2 | 2 | 65.66 | n.d | 21.47 | 0.27 | n.d | n.d | 2.49 | 8.13 | 2.3 | 0.41 | 100.79 | 26 |
|  |  |  | St. Dev. |  |  | 0.49 |  | 0.23 | 0 |  |  | 0.57 | 0.37 | 0.02 | 0.06 | 1.03 |  |
|  | IA | 61 | 13-L61-15 | (An) | 5 (10) | 67.6 | 0.02 | 17.59 | 0.5 | 0 | 0 | 0.2 | 7.54 | 5.65 | 0.13 | 99.22 | 28 |
|  |  |  | St. Dev. |  |  | 0.48 | 0.03 | 0.79 | 0.19 | 0.01 | 0 | 0.14 | 0.21 | 0.34 | 0.09 | 0.75 |  |
|  | IA | 61 | 13-L61-15 | (Plag) | 10 (18) | 50.58 | 0.05 | 28.05 | 0.57 | 0.01 | 0.17 | 13.5 | 3.64 | 0.1 | 0.01 | 96.68 | 28 |
|  |  |  | St. Dev. |  |  | 4.28 | 0.03 | 2.35 | 0.15 | 0.01 | 0.05 | 3.4 | 1.99 | 0.02 | 0.01 | 0.54 |  |
|  | IA | 63 | 63-1-6.8 |  | 4 | 65.65 | n.d | 20.5 | 0.28 | n.d | n.d | 1.04 | 7.93 | 4.52 | 0.25 | 100.37 | This study |
|  |  |  | St. Dev. |  |  | 1.5 |  | 1.14 | 0.06 |  |  | 1.25 | 0.32 | 1.53 | 0.08 | 0.74 |  |
|  | IB | 64 | f |  | 9 | 66.04 | n.d | 19.33 | 0.34 | n.d | n.d | 0.4 | 8.22 | 4.42 | 0.21 | 99.06 | 76 |
|  |  |  | St. Dev. |  |  | 0.83 |  | 0.37 | 0.11 |  |  | 0.19 | 0.57 | 0.64 | 0.09 | 0.93 |  |
|  | IB | 63 | 63-1-15.7 |  | 11 | 65.92 | n.d | 20.51 | 0.32 | n.d | n.d | 1.07 | 8.5 | 3.93 | 0.26 | 100.52 | This study |
|  |  |  | St. Dev. |  |  | 2.06 |  | 1.54 | 0.06 |  |  | 1.96 | 0.67 | 1.35 | 0.14 | 0.72 |  |
|  | IB | 64 | 64-3-9.1 |  | 8 | 66.7 | n.d | 19.2 | 0.31 | n.d | n.d | 0.32 | 8.49 | 4.18 | 0.28 | 99.47 | This study |
|  |  |  | St. Dev. |  |  | 0.59 |  | 0.14 | 0.05 |  |  | 0.12 | 0.44 | 0.71 | 0.12 | 0.73 |  |
|  | IC | 64 | f |  | 24 | 66.29 | n.d | 20.89 | 0.24 | n.d | n.d | 1.22 | 9.31 | 2.87 | 0.35 | 101.28 | 76 |
|  |  |  | St. Dev. |  |  | 0.52 |  | 0.33 | 0.03 |  |  | 0.3 | 0.21 | 0.32 | 0.08 | 0.46 |  |
|  | ICe | 66e | P8-19 |  | 27 | 64.15 | 0.03 | 20.3 | 0.27 | 0 | 0.01 | 1.37 | 9.24 | 2.73 | 0.41 | 98.51 | 28 |
|  |  |  | St. Dev. |  |  | 1.37 | 0.02 | 0.59 | 0.04 | 0.01 | 0.01 | 0.59 | 0.22 | 0.68 | 0.11 | 1.14 |  |
|  | IC | 63 | 63-1-18.0 |  | 15 | 63.47 | n.d | 21.9 | 0.3 | n.d | n.d | 2.38 | 8.55 | 2.22 | 0.38 | 99.19 | This study |
|  |  |  | St. Dev. |  |  | 2.08 |  | 1.19 | 0.07 |  |  | 1.57 | 0.6 | 0.97 | 0.1 | 0.97 |  |
|  | ID | 64 |  |  | 30 | 64.27 | n.d | 23 | 0.21 | n.d | n.d | 4.4 | 7.61 | 0.88 | 0.22 | 100.67 | 76 |
|  |  |  | St. Dev. |  |  | 1.75 |  | 1.07 | 0.07 |  |  | 1.18 | 0.51 | 0.44 | 0.14 | 0.62 |  |
|  | IDe | 66e | P8-21 | K2, K3 | 2 | 63.29 | 0.04 | 19.46 | 0.5 | 0 | 0 | 0.77 | 8.79 | 3.95 | 0.3 | 97.08 | 28 |
|  | ID | 64 | 64-3-9.9 |  | 6 | 65.07 | n.d | 20.58 | 0.27 | n.d | n.d | 1.75 | 8.76 | 2.4 | 0.34 | 99.17 | This study |
|  |  |  | St. Dev. |  |  | 1.09 |  | 0.87 | 0.12 |  |  | 1.04 | 0.3 | 1.03 | 0.18 | 0.43 |  |
|  | Ng'eju | 64 | 02-T34 |  | 11 | 59.65 | n.d | 25.24 | 0.25 | n.d | n.d | 6.59 | 7.08 | 0.66 | 0.13 | 99.74 | 26 |
|  |  |  | St. Dev. |  |  | 1.12 |  | 0.62 | 0.09 |  |  | 0.66 | 0.48 | 0.11 | 0.05 | 0.94 |  |
|  | Ng'eju | 63 | 63-1-20.5 |  | 16 | 61.11 | n.d | 23.14 | 0.27 | n.d | n.d | 5.36 | 7.9 | 1.07 | 0.24 | 99.11 | This study |
|  |  |  | St. Dev. |  |  | 3.16 |  | 1.43 | 0.07 |  |  | 2.33 | 0.99 | 0.73 | 0.14 | 1.36 |  |

**Table 1 (continued)**

| Mineral | Tuff | Loc[b] | Sample | pop | n (meas.) | SiO$_2$ | TiO$_2$ | Al$_2$O$_3$ | FeO | MnO | MgO | CaO | Na$_2$O | K$_2$O | B$_2$O | SUM | Reference |
|---|---|---|---|---|---|---|---|---|---|---|---|---|---|---|---|---|---|
| Titanomagnetite | IB | 64 | | | 16 | 0.11 | 24.74 | 0.48 | 65.99 | 1.84 | 0.52 | 0.02 | n.d | n.d | n.d | 93.86 | 76 |
| | | | St. Dev. | | | 0.07 | 0.95 | 0.1 | 1.41 | 0.11 | 0.05 | 0.05 | | | | 1.11 | |
| | IB | 63 | 63-1-15.7 | | 12 | n.d | 21.34 | 1.4 | 66.33 | 1.55 | 1.6 | n.d | n.d | n.d | n.d | 92.24 | This study |
| | | | St. Dev. | | | | 5.75 | 1.3 | 4.12 | 0.48 | 0.48 | 2.21 | | | | 1.73 | |
| | IB | 64 | 64-3-9.1 | | 10 | n.d | 24.98 | 1.06 | 66.56 | 1.55 | 0.21 | 0.03 | n.d | n.d | n.d | 94.40 | This study |
| | | | St. Dev. f | | | | 9.22 | 0.79 | 8.41 | 0.40 | 0.23 | 0.01 | | | | 1.39 | |
| | IC | 64 | | | 15 | 0.15 | 20.34 | 0.98 | 67.24 | 1.67 | 0.7 | 0.01 | n.d | n.d | n.d | 92.19 | 76 |
| | | | St. Dev. | | | 0.08 | 1.69 | 0.2 | 2.04 | 0.18 | 0.15 | 0.03 | | | | 1.1 | |
| | IC | 63 | 63-1-18.0 | | 13 | n.d | 20.73 | 1.17 | 67.49 | 1.47 | 0.89 | 0.03 | n.d | n.d | n.d | 91.75 | This study |
| | | | St. Dev. | | | | 1.75 | 0.27 | 2.36 | 0.1 | 0.43 | | | | | 2.32 | |
| | ID | 64 | 64-3-9.9 | | 15 | 0.08 | 25.17 | 1.44 | 64.27 | 1.36 | 1.19 | 0.15 | n.d | n.d | n.d | 94.92 | 76 |
| | | | St. Dev. | | | 0.03 | 0.67 | 0.17 | 1.57 | 0.06 | 0.37 | 0.13 | | | | 1.41 | |
| | ID | 64 | | | 11 | n.d | 26.15 | 1.59 | 64.81 | 1.4 | 0.18 | 0.03 | n.d | n.d | n.d | 94.17 | This study |
| | | | St. Dev. | | | | 0.82 | 0.29 | 1.71 | 0.12 | 0.08 | 0.02 | | | | 1.28 | |

a All concentrations in wt% oxide
b Locality numbers after ref. 20
c pop = population, n = number of grains analyzed, meas. = number of measurements
d Tuffs identified based on Principal Component Analysis
e Tuffs identified based on Discriminant Function Analysis of ref. 25
f Geochemical type localities based on ref. 23

sinuous meandering river flowing northwest, as revealed in trench no. 7: the Naabi ignimbrite was the result of a high energy, catastrophic volcanic eruption and associated debris flow that drastically reshaped the landscape. After this, the volcanic events were less impactful and stable environments such as river channels and floodplains were able to develop. In a palaeogeographic setting where the quarzitic basement cropped out from the extruded pyroclastic flow and a fresh water channel ran through the distal part of a volcaniclastic fan, hominins periodically used the inselberg's foothill, within short distance of both materials for tools as well as water sources from the floodplain. The floral context is established by phytoliths (Figs. 5a, 6 and Supplementary Fig. 5), in which several fern types[35–37] dominate assemblages (Fig. 6a, d, e), suggesting the existence of a fern meadow with minor woody growth and grasses; thus, pioneering bracken ferns facilitated the re-establishment of woody and grassy communities after a decrease in destructive volcanic events[38].

During deposition of the CFCTcz (2.015 ± 0.006 Ma)[25] Ma, rhizoliths in fine-grained units represent fluvial floodplain palaeosols in trench no.5 (Figs. 1, 2c, and 5). Mass flows did not allow prolonged quiescence, and laharic inundation buried synchronous woodland landscapes and Oldowan remains consisting of 12 lithics excavated 1 km west by other teams[39]. Microcharcoals from wildfires peaked (16,830 particles/cm$^3$) (range through stratigraphic sequence = 72–16,830/cm$^3$, median = 418/cm$^3$) concomitantly with grass opal increase (Supplementary Fig. 5a, b). Great diversity in phytolith morphotypes denotes markedly uneven communities and quick shifts in plant dominance (Supplementary Fig. 5d, e).

Three additional Oldowan assemblages (~2 Ma) were found at the contact between the Ngorongoro Formation and Bed I, in association with a sinuous channel flowing east, and preserved 4.6 m below Tuff IA in trench no. 5 (Figs. 1, 2f, g, 3, and 5). The reconstructed environment is a short woodland mosaic as indicated by multiple proxies. For instance, plant wax isotopes, using the weighted average δ$^{13}$C of C$_{27}$–C$_{33}$ for each individual sample and plotted using a sine-squared mixing model with end-member values of −30‰ (pure C$_3$) and −19‰ (pure C$_4$), reveal a depositional context of mosaics (Fig. 7a). One sample illustrates a C$_3$-dominated environment likely in response to increased rainfall (δ$^{13}$C: −27.9‰), whereas the other samples exhibit δ$^{13}$C values more indicative of mixed environments, ranging from δ$^{13}$C −22.6‰ to δ$^{13}$C −24.6‰. This δ$^{13}$C distribution for Ewass Oldupa is analogous to East African *Acacia-Commiphora* woodlands today. Another line of evidence indicative of woodland mosaics is stable isotopes from dental enamel: the entire analysed assemblage represents pre-Tuff IA times (δ$^{13}$C: −7.2‰ to 1.6‰), pointing to herbivores consuming a mixture of C$_3$ and C$_4$ plants (Fig. 8). Although δ$^{18}$O measurements can be influenced by a number of factors including precipitation, temperature, rainfall source, and physiology, the range obtained (δ$^{18}$O: −4.7‰ to −0.2‰) is consistent with animals obtaining their water from a similar source and no clear distinctions between taxa are obvious (Fig. 8, Supplementary Fig. 6, and Supplementary Table 3). The range for δ$^{18}$O could not be resolved to obligate or non-obligate drinking status. The enamel isotopic profile from Ewass Oldupa, in lowermost Bed I, resembles that from the younger woodlands in Bed II (Fig. 8), but contrasts with the open grassland profile prevailing in penecontemporaneous localities from western Kenya. Additional supporting evidence for a reconstructed palaeoenvironment of woodland/grassland mosaics is shown by phytoliths (Table 2, Figs. 5, 6, and Supplementary Fig. 5b), characterized by fluctuating woody dicot and grass outputs.

A fifth Oldowan assemblage includes stone tools and bones 4 m below Tuff IA (~2 Ma) on a transgressive lakeshore representing the earliest and furthest western expansion of palaeolake Oldupai,

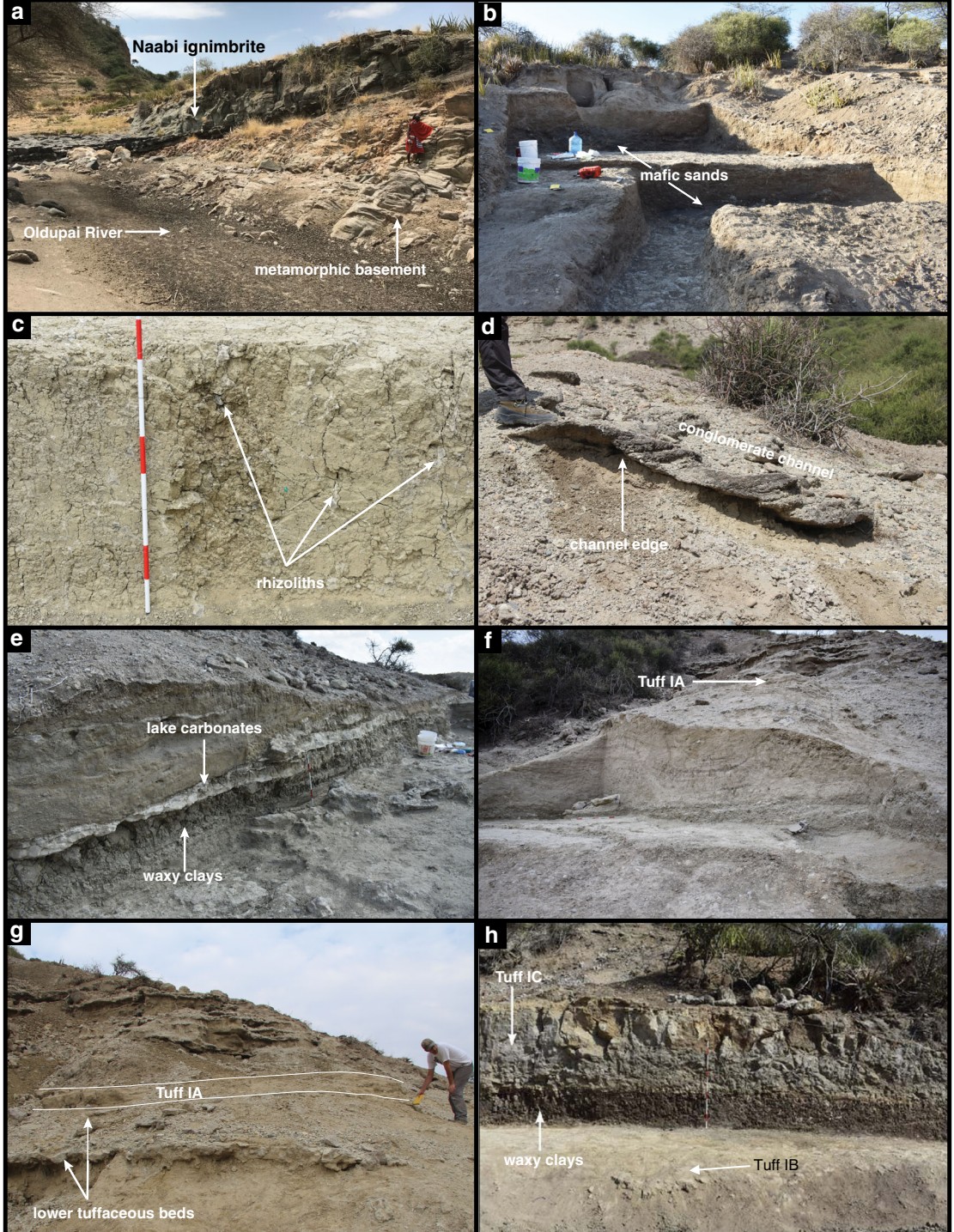

**Fig. 2 Key stratigraphic horizons. a** Ngorongoro formation: Naabi ignimbrite overlying metamorphic basement along the Oldupai River. **b** Ngorongoro Formation: Mafic sands immediately above the Naabi in trench 7. **c** Ngorongoro formation: CFCT compositional zone, trench 5, characterized by abundant rhizoliths. **d** Lower Bed I: conglomeratic channel fill deposit with trough-cross stratification dipping to the east. **e** Lower Bed I: waxy claystones from trench 3 capped by carbonate beds. **f** Lower Bed I: multi-storey fluvial channel belt deposits in trench 2, several meters beneath Tuff IA. (Metric pole on the ground for scale.) **g** Lower Bed I: Ewass Oldupa exposure of Tuff IA and underlying thin tuffaceous beds. **h** Upper Bed I: thick waxy claystone unit capping Tuff IB, and underlying Tuff IC. See Fig. 5 for further stratigraphic information.

as exposed in trench no. 3 (Figs. 1, 2e, 3, and 5). Taxonomically diverse fauna without hominin modification include Antelopini, Bovini, Hippotragini, and Tragelaphini (Supplementary Table 4, Supplementary Note 1), showing mixed habitats, in addition to water-dependent taxa. Phytoliths show a floristic tandem of woodlands and palms that epitomises the Oldowan lake margin

occupations characteristic of Upper Bed I[40–43] (Figs. 5–6, Table 2), representing proximity of fresh water and preferred hominin sources of raw materials. Further, the Oldowan stone tools from this horizon (Fig. 4e, f) are the earliest lithic evidence deposited in a waxy clay from a saline/alkaline lacustrine environment anywhere.

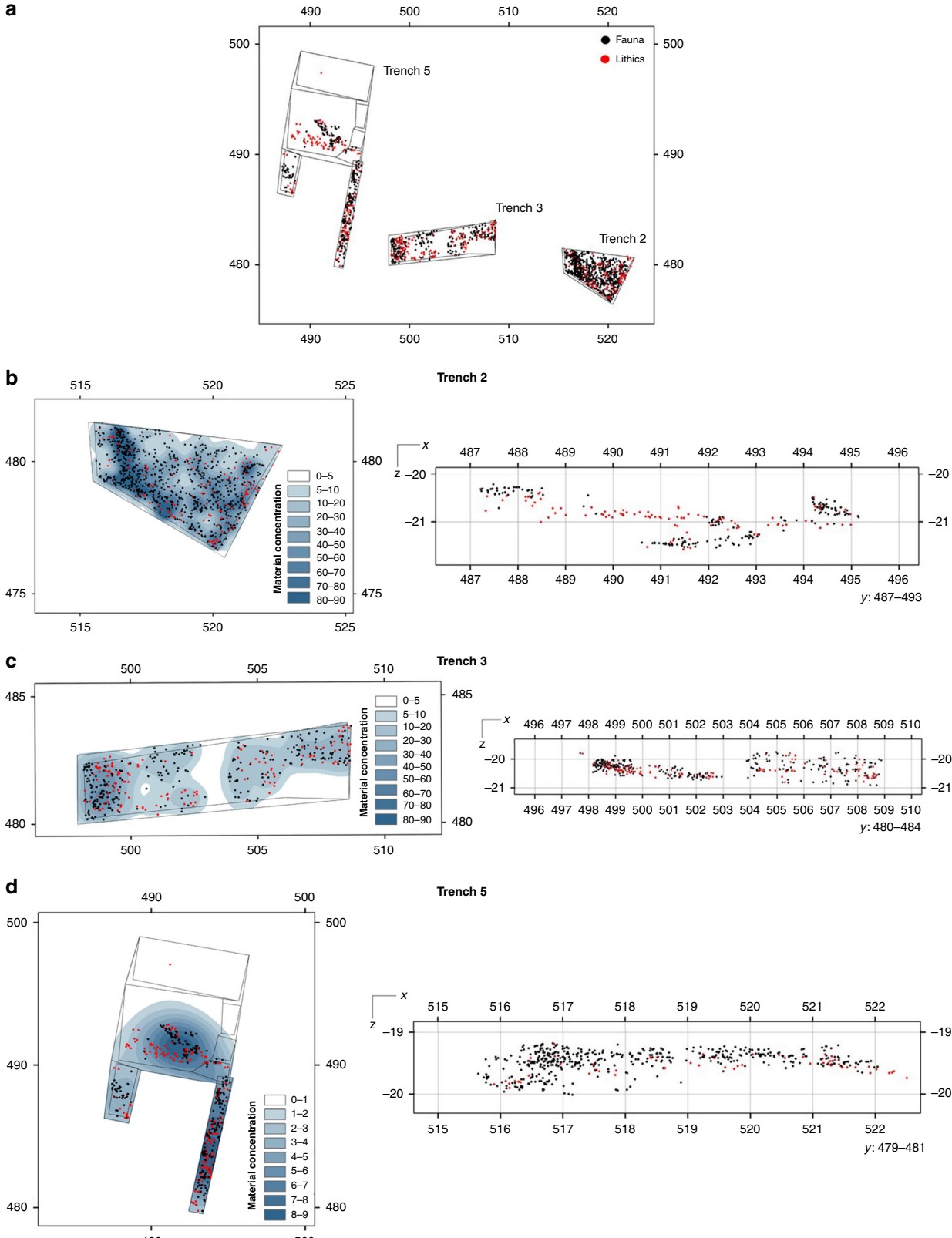

**Fig. 3 Mapping of excavated materials. a** Plotting of excavated lithics and fossil specimens. **b–d** Right: vertical projections of stone artifacts and bones show discrete archaeological horizons (axis units are in meters). Left: Kernel density analysis shows spatial variations in the accumulation of archaeological materials (densest in darkest blue). The stratigraphic position of trenches 2, 3, and 5 within the overall column is presented in Fig. 5.

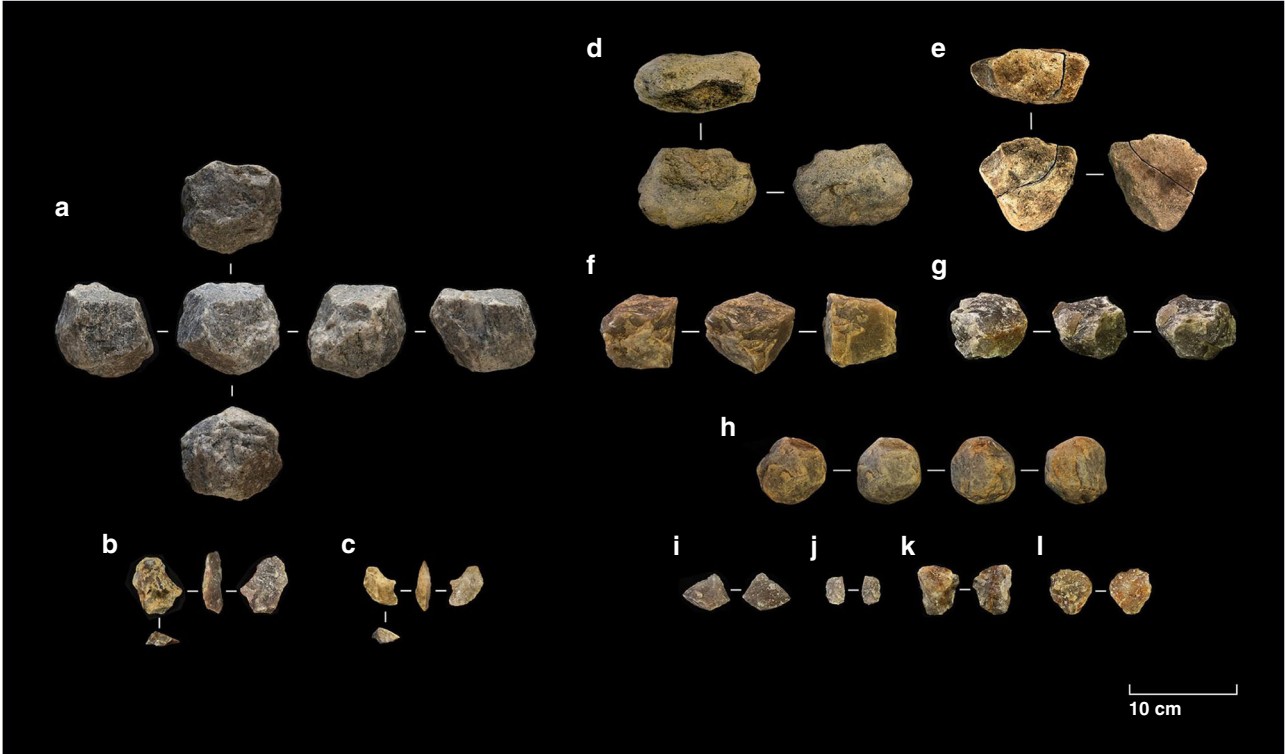

**Fig. 4 Selection of stone tools from Ewass Oldupa.** Ngorongoro Formation: earliest Oldowan, post Naabi, Trench 7: **a** Quartzite multipolar-multifacial core. **b**–**c** Quartzite flakes. Ngorongoro Formation, Contact CFCT compositional zone/Bottom of Bed I, Trench 5: **d** Ignimbrite chopping-tool. **e** Ignimbrite chopper. **f** Quartzite unipolar longitudinal core. **g** Quartzite multipolar-multifacial core. Lower Bed I: earliest lake expansion, Trench 3: **h** Quartzite spheroid. **i** Quartzite flakes. Lower Bed I: prograding fluvial system below Tuff IA, Trench 2: **j**–**l** Quartzite flakes.

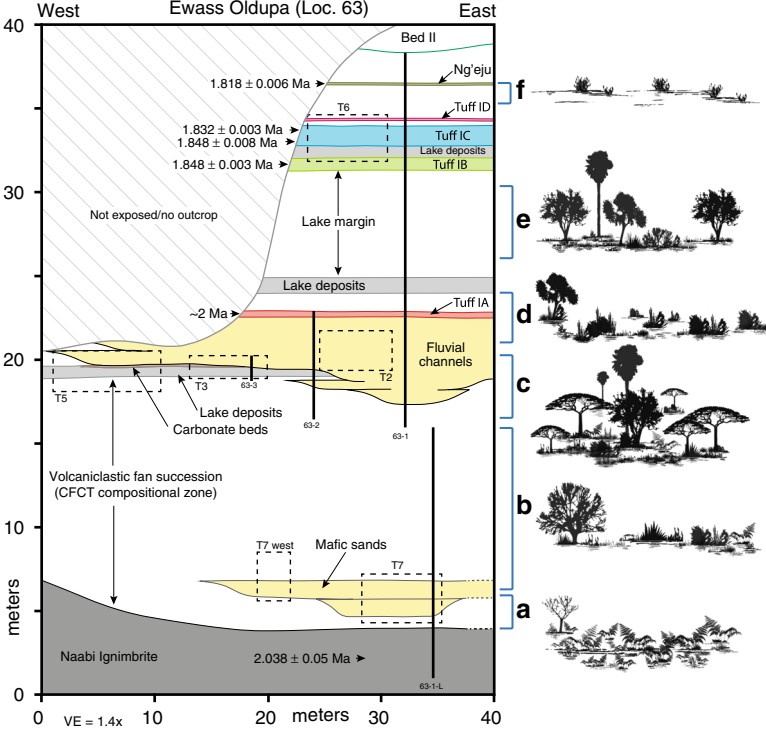

**Fig. 5 Outcrop geometry, stratigraphic architecture, and idealized vegetation at Ewass Oldupa.** Location of measured stratigraphic sections and excavation trenches. The details of section 63-1-L and 63-1 are presented in Fig. 1b. Artist rendering of plant communities over time: **a** Post-eruptive, fern meadow. **b** CFCT mosaics. **c** Woodland with palms and ferns. **d** Grasslands coeval with Tuff IA. **e** Open woodland. **f** Asteraceae-dominated scrub.

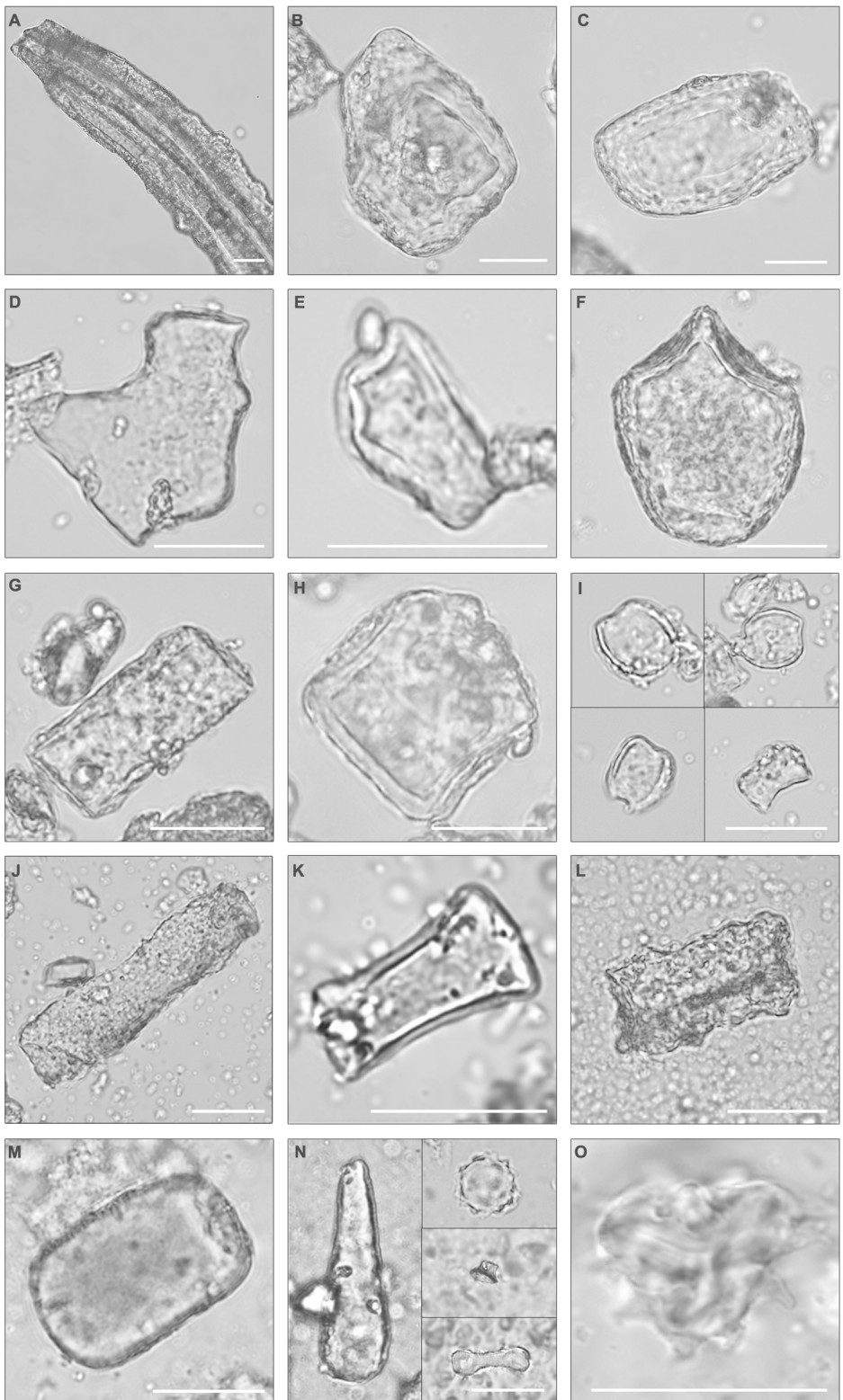

**Fig. 6 Photomicrographs of selected phytoliths and pollen from Bed I. a** Large tabular sulcate cf. Pteridaceae. **b**, **c** Blocky phytoliths from woody dicots. **d** Epidermal piece cf. Pteridaceae. **e** Tabular bifid cf. Pteridaceae (e.g. *Cyrtomium*). **f** Bulliform, Poaceae. **g** Tabular. **h** Shield cf. Solanaceae. **i** Saddle, short, Chloridoideae. **j** Tabular scrobiculate. **k** Tabular strangulated cf. *Salvadora*. **l** Tabular sinuate from woody dicot. **m** Blocky from woody dicot. **n** Left. Clavate from woody dicot. Right. Upper: globular echinate, Arecaceae. Middle: Tower, Poaceae. Bottom: Bilobate, long, convex, Poaceae. **o** Asteraceae pollen from Upper Bed I, immediately underneath the Ng'eju Tuff. Pollen concentration = 2649; C3 Ret. Lamiaceae/Convolvulaceae, 101; Asteraceae, 72; Rutaceae, 2. All scale bars = 25 μm. .

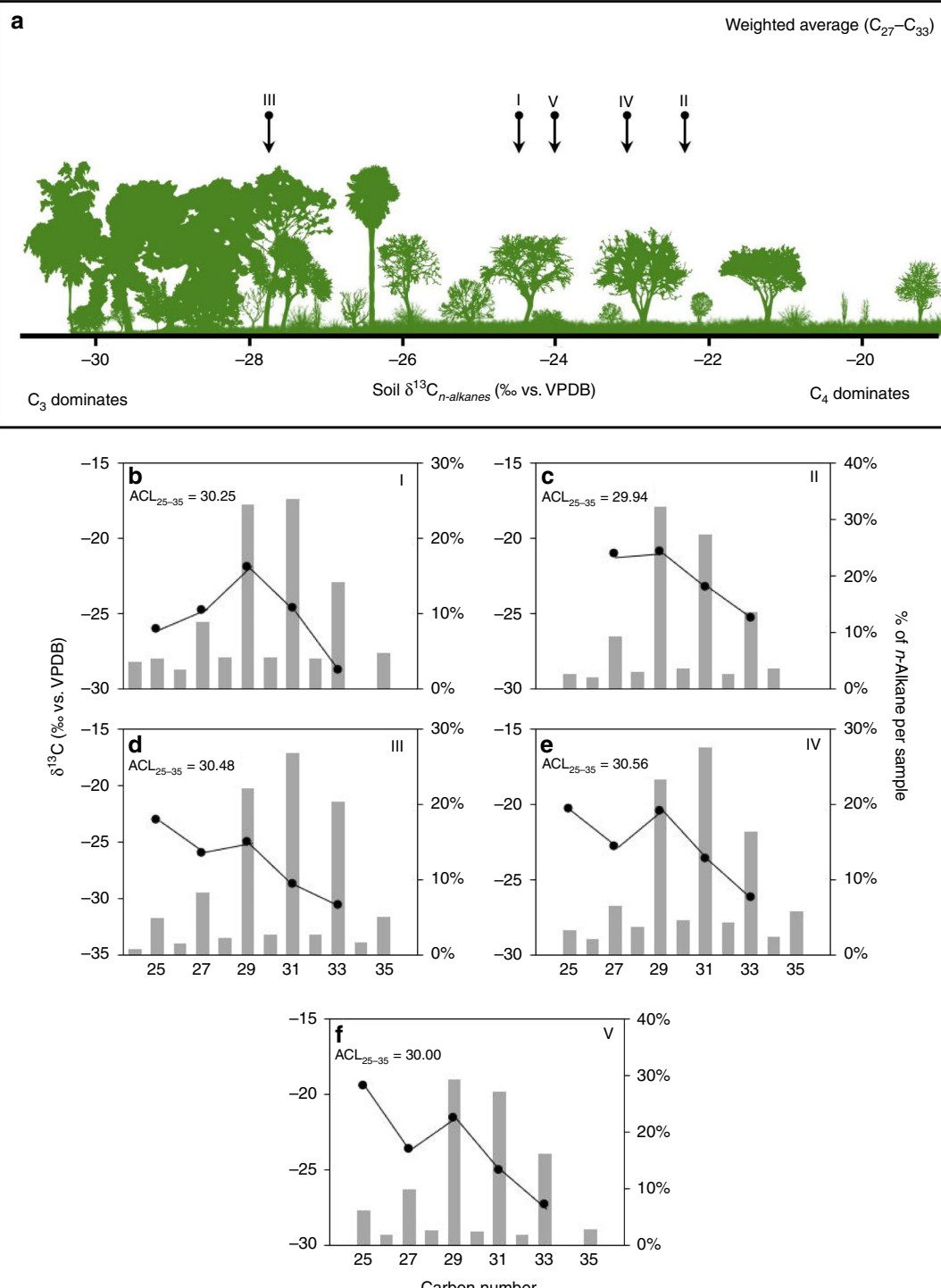

**Fig. 7 Plant wax biomarkers. a** Plant landscape reconstruction using $\delta^{13}$C of the weighted mean average of the $C_{27}$–$C_{33}$ $n$-alkanes. A sine-squared mixing model with end-member values of −30‰ (for pure $C_3$) and −19‰ (for pure $C_4$) was used to visualize plant ecology during sample deposition. The $\delta^{13}$C of $n$-alkanes vary between −29‰ and −39‰ in extant $C_3$ plants and −14‰ and −26‰ in $C_4$ vegetation. VPDB Vienna Pee-Dee Belemnite. **b**–**f** Graphs show relative abundance of each $n$-alkane compound (grey bars), their $\delta^{13}$C values (black circles), and average chain length (ACL) of $C_{25}$–$C_{35}$ carbon homologue. Roman numerals indicate provenance in the stratigraphic column from bottom to top, and these are individual samples, not composite. The placement of each sample along the soil/plant gradient is based on the obtained weighted average for each sample.

A sixth Oldowan cluster was located near a meandering stream within a multi-storey channel belt prograding into the lake, positioned 3 m beneath Tuff IA (~2 Ma) and uncovered in trench no. 2 (Figs. 1, 2f, 3, and 5). Fauna does not show marks of hominin processing (Supplementary Note 1). It comprises bovids, suids, equids, large mammals, and Plio-Pleistocene cercopithecines that feed in grassy environments (cf. *Theropithecus oswaldi*)[44,45] as well as perennial water indicators (*Chelonia*, *Crocodylus*, and *Hippopotamus*) (Supplementary Table 4). Phytoliths show woody dicots and grasses booming and busting cyclically[46], while grassland encroachment on woodlands occurred twice (Supplementary Fig. 5b).

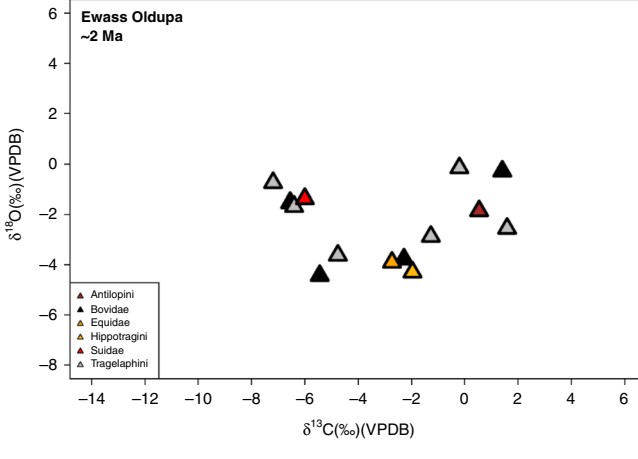

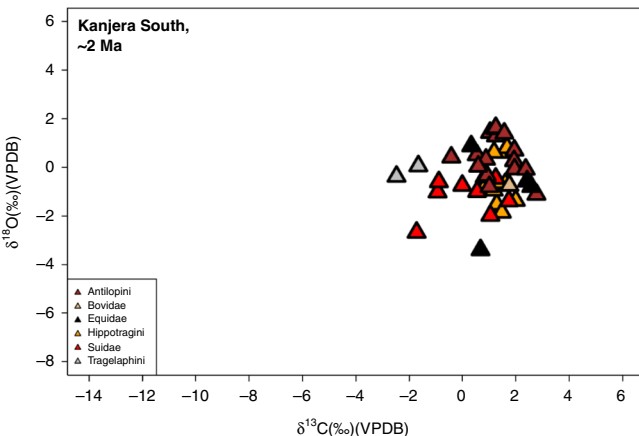

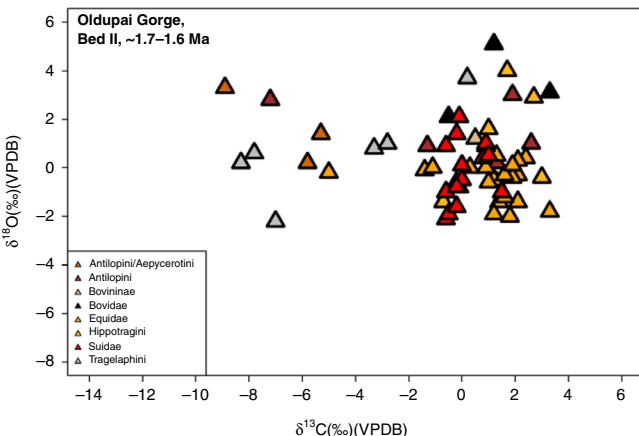

**Fig. 8 Stable carbon (δ13C) and oxygen (δ18O) measurements of animal teeth from Ewass Oldupa (below Tuff IA) compared to contemporaneous and younger fossil datasets[14,75].** A Mann–Whitney–Wilcoxon test shows significant difference in δ13C [$W = 215$, $<0.05(0.004)$]. There is also δ18O distinction: [$W = 92$, $<0.05(0.000)$]. Overall, the results suggest that the period 2.0–1.9 Ma was wetter than 1.8–1.6 Ma. Note: only data from the same families of taxa have been included in the comparison.

Isotopic *n*-alkane (C$_{29}$ C$_{31}$ C$_{33}$) values for Upper Bed I from regressive lakeshores and small streams between Tuff IA-IB show open woodland with mixed C$_3$–C$_4$ habitats (Figs. 1 and 7a, d–f) in a stratigraphic position equivalent to that of the oldest *Homo habilis* in the Eastern basin[30] (OH 24). The succession of Oldowan occupations ends with artefacts from a lakeshore context sandwiched between Tuff IB and IC (1.848/1.832 Ma), in a position and sedimentary environment that on the Western

lakeshore is a counterpart to Level 22 Zinj[23,24,41,47], while still 2 m below OH 65[21] (Fig. 1), as uncovered in trench no. 6. Pollen grains express a treeless, grassless, hyper-xeric steppe dominated by Asteraceae in cohort with Lamiaceae/Convolvulaceae and Rutaceae scrub (Fig. 6o) immediately underneath the Ng'eju Tuff, correlative with a mudstone/calcareous horizon over OH 65[22,48].

## Discussion

Ewass Oldupa is a high-resolution, multi-episode, early Oldowan stratified site that precedes and straddles Bed I. This site holds Oldupai's oldest evidence for hominin cultural remains associated with habitats from post volcanic disturbance and thus early examples of adaptation to major geomorphic and ecological transformations. The earliest Oldowan tools appear immediately after the deposition of the Naabi ignimbrite, and they are associated with a stacked fluvial series composed of mafic tephra (Fig. 1). Therein, cross-stratified sandstone and an overall upward fining capped by siltstone shows that humans provisioned near meandering rivers. This episodic exploitation of fluvial environments precedes that of the unstable surfaces characteristic of younger CFCT horizons[39]. Large lithic and faunal assemblages from the contact between the topmost CFCT and lowermost Bed I, together with a wide range of vegetation proxies, mark the onset of lacustrine occupation in the basin, which precedes the emplacement of Tuff IA. The Oldowan toolkits from this time intersect earlier and contemporaneous technologies from Ethiopia and Kenya, while advancing tool types such as the spheroid commonly seen in younger assemblages[24]. If these stone tools drove provisioning amidst unpredictable environments, it was via a generalist strategy[49,50] for a variety of tasks that did not include defleshing animal carcasses, as shown by a lack of human modified fauna.

In conclusion, the earliest Oldupai hominins occupied heterogeneous unstable environments. Sedimentary, chemical, and vegetation data indicate that early Oldowan hominins pioneered a rapid occupation of geo-settings undergoing drastic changes in hydrological resource distribution and structure, and supporting uneven floras. The evidence uncovered is of periodic, recurrent land use across a subset of environments punctuated with times when there is no evidence of hominin activity. Our work reveals early Oldowan stone tools in diverse physical environments, with indications that these environments both changed significantly over space and time. In addition, Oldowan groups utilised disturbance environments—a finding that is unique for this period and depicts complex behavior among early Pleistocene hominins. The ability to exploit diverse biomes enabled hominins to expand beyond Africa. This behavioral flexibility and adaptive suite materialized in the absence of controlled fire, amid large predators, and can be interpreted as a predictor of the invasiveness that facilitated early *Homo*'s global dispersal.

## Methods

**Biomarkers**. Dry, homogenized sediment was PSE extracted (Büchi Speed Extractor E-916) in three 10 min cycles in 9:1 (v:v) DCM:MeOH at 100 °C and 103 bar/1500 psi. Neutral, non-polar hydrocarbons were separated from total lipid extracts by Aminopropyl column chromatography using 2:1 DCM:IPOH. Normal (*n*-) alkanes were separated from the neutral fraction using silver nitrate-infused silica gel column chromatography using Hexane. Gas chromatography (GC) analysis was performed using an 7890B (Agilent) and HP-5 column (Agilent; 30 m length, 0.25 mm i.d., and 0.25 μm film) with a 5977 A MSD. Samples were injected at 250 °C in splitless mode and the oven temperature programmed from 60 °C (1 min hold) to 150 °C at 10 °C/minute, then to 320 °C at 6 °C/minute (10 min hold). The Mass Spectrometry (MS) source was operated at 230 °C with 70 eV and a fullscan rate of m/z 50–550. Plant wax *n*-alkanes were identified by comparing mass spectra and retention time with an external standard mixture (C$_{21}$–C$_{40}$).

Isotope-ratio MS Compound-specific δ13C values were analysed by 7890B GC (Agilent) with a HP-5 capillary column (Agilent) operated in split mode coupled to

**Table 2 Phytolith counts per morphotype.**

| Sample | Poaceae, short cell | | | | | | | | | | Woody Dicot | | | | | | | | | | | | | | | | | | | | | | |
| | Lobate | | | | | Rondel | | Saddle | | Sq | Blocky | | Cylindrical | | | | | | | Fusiform | | Sclereid/Clavate | | Spherical | | | | | | | | |
| | BLC | BLF | BSCave | BSCvex | BSF | T | W | L | Sh | Sq | B | BC | C | CA | CC | CCr | CP | CSc | Csi | F | G | C | S | GE | GF | GG | GP | GGO | GT | G | HP | OG |
|---|---|---|---|---|---|---|---|---|---|---|---|---|---|---|---|---|---|---|---|---|---|---|---|---|---|---|---|---|---|---|---|---|
| 1 | 0 | 0 | 0 | 0 | 0 | 0 | 0 | 0 | 4 | 1 | 5 | 0 | 0 | 0 | 0 | 0 | 0 | 3 | 0 | 1 | 0 | 0 | 0 | 0 | 0 | 0 | 0 | 0 | 0 | 3 | 0 | 0 |
| 2 | 0 | 0 | 0 | 0 | 0 | 0 | 0 | 0 | 3 | 0 | 15 | 0 | 0 | 0 | 0 | 0 | 0 | 0 | 0 | 3 | 1 | 0 | 12 | 0 | 0 | 3 | 0 | 0 | 1 | 3 | 1 | 0 |
| 3 | 0 | 0 | 0 | 0 | 0 | 0 | 0 | 0 | 0 | 2 | 14 | 0 | 0 | 0 | 0 | 0 | 0 | 1 | 0 | 0 | 0 | 0 | 12 | 0 | 0 | 3 | 0 | 0 | 0 | 5 | 0 | 0 |
| 4 | 6 | 0 | 0 | 0 | 0 | 1 | 0 | 2 | 17 | 0 | 9 | 1 | 0 | 2 | 2 | 0 | 18 | 0 | 0 | 0 | 0 | 0 | 2 | 0 | 1 | 0 | 2 | 0 | 2 | 5 | 0 | 0 |
| 5 | 0 | 1 | 5 | 2 | 0 | 3 | 0 | 0 | 20 | 7 | 9 | 0 | 0 | 0 | 0 | 0 | 5 | 5 | 0 | 2 | 0 | 0 | 2 | 1 | 0 | 2 | 1 | 0 | 0 | 4 | 0 | 0 |
| 6 | 0 | 0 | 0 | 0 | 0 | 0 | 0 | 0 | 0 | 0 | 15 | 0 | 0 | 0 | 4 | 0 | 7 | 7 | 5 | 2 | 0 | 0 | 3 | 0 | 0 | 5 | 0 | 0 | 0 | 4 | 0 | 0 |
| 7 | 0 | 0 | 0 | 0 | 0 | 0 | 0 | 0 | 0 | 0 | 14 | 0 | 27 | 0 | 0 | 1 | 0 | 0 | 0 | 0 | 0 | 0 | 0 | 0 | 0 | 1 | 0 | 0 | 0 | 81 | 0 | 0 |
| 8 | 0 | 0 | 0 | 0 | 0 | 0 | 0 | 0 | 0 | 0 | 2 | 0 | 0 | 0 | 0 | 0 | 0 | 0 | 0 | 2 | 0 | 0 | 0 | 0 | 0 | 3 | 0 | 0 | 0 | 71 | 0 | 0 |
| 9 | 0 | 0 | 1 | 0 | 1 | 0 | 0 | 0 | 0 | 0 | 18 | 0 | 0 | 0 | 0 | 0 | 1 | 0 | 0 | 0 | 1 | 2 | 0 | 1 | 0 | 0 | 0 | 0 | 0 | 13 | 0 | 0 |
| 10 | 0 | 0 | 0 | 0 | 0 | 0 | 0 | 0 | 0 | 0 | 0 | 0 | 0 | 0 | 0 | 0 | 0 | 4 | 0 | 1 | 0 | 2 | 0 | 6 | 0 | 0 | 0 | 0 | 0 | 5 | 1 | 0 |
| 11 | 0 | 1 | 0 | 0 | 0 | 0 | 0 | 0 | 0 | 0 | 7 | 0 | 0 | 0 | 0 | 0 | 0 | 0 | 0 | 0 | 0 | 0 | 0 | 2 | 0 | 0 | 0 | 0 | 0 | 1 | 1 | 0 |
| 12 | 0 | 0 | 1 | 0 | 0 | 2 | 3 | 0 | 1 | 1 | 6 | 0 | 0 | 0 | 0 | 0 | 6 | 0 | 0 | 0 | 1 | 0 | 0 | 0 | 7 | 0 | 0 | 0 | 0 | 7 | 0 | 0 |
| 13 | 0 | 0 | 2 | 0 | 0 | 1 | 1 | 0 | 0 | 0 | 1 | 0 | 0 | 0 | 0 | 0 | 0 | 4 | 5 | 0 | 0 | 0 | 0 | 2 | 43 | 0 | 0 | 0 | 0 | 11 | 0 | 0 |
| 14 | 0 | 0 | 0 | 0 | 0 | 0 | 0 | 0 | 0 | 0 | 4 | 0 | 0 | 0 | 0 | 0 | 10 | 0 | 0 | 2 | 0 | 0 | 0 | 5 | 0 | 4 | 0 | 0 | 0 | 26 | 0 | 0 |
| 15 | 0 | 0 | 0 | 0 | 0 | 0 | 0 | 0 | 0 | 0 | 1 | 0 | 0 | 0 | 0 | 0 | 0 | 0 | 0 | 1 | 1 | 0 | 0 | 2 | 0 | 3 | 0 | 0 | 0 | 38 | 0 | 0 |
| 16 | 0 | 0 | 0 | 0 | 0 | 0 | 0 | 0 | 0 | 0 | 6 | 0 | 4 | 0 | 0 | 0 | 0 | 0 | 0 | 0 | 0 | 0 | 0 | 4 | 0 | 2 | 0 | 3 | 3 | 3 | 4 | 0 |
| 17 | 0 | 0 | 0 | 0 | 0 | 0 | 0 | 0 | 1 | 0 | 5 | 0 | 3 | 0 | 0 | 0 | 5 | 0 | 0 | 3 | 0 | 0 | 3 | 10 | 0 | 6 | 0 | 0 | 8 | 16 | 1 | 2 |
| 18 | 0 | 0 | 2 | 0 | 0 | 0 | 0 | 0 | 9 | 0 | 7 | 0 | 0 | 0 | 0 | 0 | 4 | 0 | 0 | 0 | 0 | 0 | 0 | 2 | 0 | 1 | 0 | 0 | 9 | 2 | 0 | 0 |
| 19 | 0 | 0 | 0 | 0 | 0 | 1 | 0 | 0 | 0 | 0 | 12 | 0 | 0 | 0 | 0 | 0 | 0 | 0 | 0 | 0 | 0 | 0 | 1 | 1 | 0 | 5 | 1 | 0 | 2 | 5 | 0 | 0 |
| 20 | 0 | 2 | 5 | 0 | 1 | 0 | 0 | 2 | 195 | 6 | 1 | 0 | 4 | 0 | 0 | 0 | 2 | 3 | 0 | 0 | 0 | 0 | 0 | 2 | 0 | 0 | 1 | 0 | 11 | 122 | 0 | 0 |
| 21 | 0 | 0 | 0 | 0 | 0 | 0 | 0 | 0 | 7 | 0 | 11 | 0 | 0 | 0 | 0 | 0 | 3 | 0 | 0 | 0 | 1 | 2 | 1 | 0 | 0 | 2 | 0 | 0 | 0 | 13 | 0 | 0 |
| 22 | 0 | 0 | 0 | 0 | 0 | 0 | 0 | 0 | 0 | 0 | 7 | 0 | 0 | 0 | 0 | 0 | 2 | 0 | 0 | 0 | 0 | 0 | 0 | 0 | 0 | 2 | 0 | 0 | 0 | 3 | 0 | 0 |
| Total | 6 | 4 | 16 | 2 | 2 | 8 | 4 | 4 | 257 | 17 | 169 | 1 | 38 | 2 | 6 | 1 | 63 | 29 | 10 | 17 | 5 | 6 | 36 | 38 | 51 | 42 | 5 | 3 | 36 | 441 | 7 | 2 |

(Abbreviations, left to right) *BLC* Bilobate, long shaft, concave ends, *BLF* Bilobate, long shaft, flat ends, *BSCave* Bilobate, short shaft, concave ends, *BSCvex* Bilobate, short shaft, convex ends, *BSF* Bilobate, short shaft, flat ends, *T* Rondel, tower, *W* Rondel, wide, *L* Saddle, long, *Sh* Saddle, short, *Sq* Saddle, squat, *B* Blocky, *BC* Blocky, cavate, *C* Cylinder, *CA* Cylinder, areolate, *CC* Cylinder, cavate, *CCr* Cylinder crenate, *CP* Cylinder, psilate, *CSc* Cylinder, scrobiculate, *CSi* Cylinder, sinuate, *F* Fusiform, *G* Guttiform, *C* Clavate, *S* Sclereid, *GE* Globular, echinate, *GF* Globular, facetate, *GG* Globular, granulate, *GP* Globular, psilate, *GGO* Globular, granulate, oblong, *GT* Globular tuberculate, *G* Globulose, *HP* Hemisphere, psilate, *OG* Oblong, granulate, *TC* Tabular, cavate, *TEl* Tabular, ellipsoidal, *TSc* Tabular, scrobiculate, *TSi* Tabular, sinuate, *TSt* Tabular, strangulated, *TSu* Tabular, sulcate, *TTL* Tabular, thick, lacunate, *TR* Tabular, ridged, *B* Bulliform, *T* Trichome, *P* Papillae, *H* Hair, *El* Epidermal, jigsaw, *V* Vessel, *T* Tabular, *TC* Tabular crenate, *TD* Tabular, dendritic, *TF* Tabular, facetate, *TP* Tabular, psilate, *TS* Tabular, striated, *TV* Tabular, velloate.

a Elementar Isoprime AnthrovisION Mass Spectrometer by an Elementar GC5 furnace system interface operating at 850 °C. Oven temperature was programmed from 60 °C (1 min hold) to 150 °C at 10 °C/minute, then to 320 °C at 6 °C/minute (10 min hold). Helium was the carrier gas with a constant flow of 1.1 mL/min. Accuracy of $\delta^{13}C$ values were evaluated against an international standard laboratory mixture (Indiana B4, Arndt Schimmelmann, University of Indiana) injected at the beginning and end of each sample set. The standard deviation of the n-alkane working standard was ≤0.5‰. Isotopic ratios are expressed as $\delta^{13}C$ values in per mil relative to the Vienna Pee Dee Belemnite (VPDB) standard.

**Energy dispersive X-ray fluorescence.** Quartzite artefacts ($n = 24$) were individually sonicated (10 min; 40 kHz) followed by chemical characterisation using EDXRF (Thermo Scientific ARL QUANT'X; end window bremsstrahlung; Rh target X-ray tube 50 W; 76 µm beryllium window; 4–50 kV; Edwards RV8 vacuum pump). Each analytical run ($n = 5$) included the USGS RGM-2 reference standard for instrument calibration. The targeted elements included eight major oxides ($SiO_2$, $TiO_2$, $Al_2O_3$, $Fe_2O_3$, MnO, MgO, CaO, and $K_2O$) and 10 trace elements (Cu, Zn, Rb, Sr, Y, Zr, Nb, Ba, Pb, and Th). X-ray intensities were automatically converted to concentration estimates (major oxides = wt %; trace elements = ppm) using a least-squares calibration line ratioed to the Compton scatter for each element based on international reference standards (AGV-2, BCR-2, BHVO-2, BIR-1a, GSP-2, JR-1, JR-2, QLO-1, RGM-2, SDC-1, STM-2, TLM-1, and W-2a). Concentration estimates were normalised according to the USGS RGM-2 reference standard using Min–Max scaling, with each value rescaled to range [0, 1]. The same analytical conditions and data treatment methods were implemented for quartzite geological samples ($n = 125$) from five outcrops, which constitute the reference collection used for comparative analyses[51,52]. Data exploration and Linear Discriminant Analysis of the normalised chemical data were performed in RStudio using 14 packages[51,52].

**Excavation.** Location of materials was recorded on a Leica TS 09 during excavation. Materials were placed within an $X$–$Y$ Cartesian system, with $Z$-values also recorded, prior to being individually labelled and bagged. Sediments from each trench were dry-sieved (0.5 cm mesh). Identification and $XYZ$ coordinates were exported to a database. This field database was the matrix for techno-typological classification, taxonomy, taphonomy, and spatial analysis. 3D-registered data was $XY$-plotted using ArcMap10.7, with $XZ$ and $YZ$ projections performed N–S and E–W. Kernel density analysis (search radius = 25 cm) was applied to $XY$ data.

**Fauna.** Elements >2 cm were studied. For taxonomic identification, we consulted the Smithsonian Institution's digital archive of mammals from East Africa[53], an osteological manual[54], an established museum reference collection directly available to us at IPHES (Tarragona, Spain), as well as specialised faunal literature on Early Pleistocene fossils[55]. Surface modifications were noted for all specimens with cortex. The specimens were then studied under a stereomicroscope (Optech HZ, 10–60x). Skeletal profiles were created using the values for the number of identified specimens (NISP). Size class classification[56–58] was applied where no higher order classification could be made. Minimum number of elements (MNE) was calculated based on NISP, with frequency, portions, size, and side under analysis being noted[59]. MNE portions were used to standardise minimum animal units (% MAU)[60]. A correlation coefficient for MAU percentage and bone mineral density[61], considering the MNE of each portion of all elements among ungulates. The age of a specimen was stated where possible, with "unknown" used otherwise. Fracture patterns for long and flat bones were noted[62].

**Mineral geochemistry.** Composition of phenocrysts was determined by electron microprobe analysis. Preliminary petrographic analysis of polished thin sections used cross-polarizing, petrographic microscope to identify phenocryst abundance, sorting, degree of alteration, and suitability for analysis. Samples were coated with carbon and analyzed using the JEOL JXA-8200 electron microprobe. Electron microprobe beam was operated at a current of 8 µA and a voltage of 15 kV and a beam diameter of 11 mm. A combination of brightness from BSE images and EDS was used to identify potential mineral grains to analyze. Where possible, a minimum of 10 grains were analyzed per sample per mineral (30-point/sample). Microprobe calibration standards for feldspar grains were orthoclase and albite, standards for augite/clinopyroxene grains were Cr-augite and hornblende, and ilmenite was used to calibrate for oxide grains. Effort was made to analyze grains that were contained within glass or within pumiceous material. Mineral grains were analysed at random to prevent bias. Wt% totals for feldspar and pyroxene analyses less than 96% or greater than 102% were excluded from the final data, as were titanomagnetite grains with totals less than 90%.

**Phytolith analysis.** Samples (3 g) were mixed with 0.1% sodium hexametaphosphate ($NaPO_3$)$_6$ and sonicated (5 min). Orbital shaking was performed overnight (200 rpm). After clay dispersal, 3 N hydrochloric (HCl) and nitric acids ($HNO_3$) plus hydrogen peroxide ($H_2O_2$) was applied. Sodium polytungstate ($3Na_2WO_4$· $9WO_3$· $H_2O$) (Poly-Gee) at specific gravity 2.4 separated out phytoliths. Rinsing and centrifugation was done at 3000 rpm for 5 min. Aliquot (0.001 g) was mounted on microscope slide with Entellan New (cover: 20 × 40 mm = inspected area).

System microscopy was at 40x (Olympus BX41, Motic BA410E). Minimum of 200 phytoliths were counted where possible, or until sample extinction. The referential included several African ecoregions[46,63–68], and past work at Oldupai Gorge[43,69]. Archaeological samples were compared to a baseline of 29 plant species and 35 soils from *Acacia-Commiphora* woodland mosaics from the study area, using general and diagnostic approaches to reconstruct physiognomy and plant cover rank through their phytolith signal[46]. Classification nomenclature followed ref. 70.

**Pollen and microcharcoal.** Protocol was adapted from ref. 71. Sample (2 mL) was mixed with 3% sodium hexametaphosphate ($NaPO_3$)$_6$ in waterbath (90 °C) for 30 min and mechanically agitated over several days. *Lycopodium* spore tables were added to calculate pollen and microcharcoal concentrations. Samples settled and decanted (<2 µm), followed by sieving (>125 µm). Separation of pollen and charcoal was done using gentle mechanical agitation over several days followed by heavy liquid separation using lithium heteropolytungstate at specific gravity 2.0. Acetolysis was achieved through 9:1 mixture of $C_4H_6O_3$ to $H_2SO_4$ and 10% HCl. The samples were mounted on microscope slide with glycerol (cover: 22 × 40 mm = inspected area). System microscopy was done at 25x, 40x, 63x. Minimum of 350 pollen grains and 200 microcharcoal were counted.

**Stable carbon and oxygen isotope analysis of faunal dental enamel.** Specimens selected all had excellent preservation conditions, and no weathered or fragmented dental specimens were considered. Teeth were cleaned using air-abrasion. Sample of enamel powder (6 mg) was taken from buccal edge using diamond-tipped burr drill. It was washed in 1.5% NaClO for 60 min, rinsed, then centrifuged. It was then washed with acetic acid (0.1 M) for 10 min and rinsed. Residue was lyophilised for 24 h, then set to react with $H_3PO_4$ and gases were measured for stable carbon and oxygen ratios using GasBench II (Thermo) coupled with Delta V Advantage MS (Thermo). Values were compared against International Standards: IAEA-603 ($\delta^{13}C$ = 2.5; $\delta^{18}O$ = −2.4); IAEA-CO-8 ($\delta^{13}C$ = −5.8; $\delta^{18}O$ = −22.7); USGS44 ($\delta^{13}C$ = −42.2), and in-house standard MERCK ($\delta^{13}C$ = −41.3; $\delta^{18}O$ = −14.4). Measurement error of MERCK standard c was ±0.1‰ for $\delta^{13}C$ and ±0.2‰ for $\delta^{18}O$. Overall precision of measurements through repeat extracts from an in-house bovid tooth enamel standard c was ±0.2‰ for $\delta^{13}C$ and ±0.3‰. Obligate or non-obligate drinking status was inferred from the range of $\delta^{18}O$ as per references[72–74].

**Stone tools.** All specimens were measured and weighed, with attributes recorded in database. Technological classes such as core, core fragment, flake, broken flake, flake fragment with a platform, percussive material, retouched piece, and tools were distinguished. Knapping methods were classified according to faciality (noting the number of surfaces exploited) and polarity (direction of exploitation from a striking platform). The following core reduction methods were observed: (i) multipolar-multifacial: cores with four or more surfaces exploited in at least four different directions; (ii) unipolar-longitudinal: cores with a single surface exploited, unidirectionally; (iii) orthogonal-bifacial: cores with two opposed surfaces knapped, unidirectionally; (iv) centripetal-bifacial: cores with two exploitation surfaces organized by a horizontal plane, each side with at least three centripetal removals; (v) centripetal-unifacial: cores with one surface exploited and at least three centripetal extractions; (vi) bipolar-on-anvil: cores with evidence of placement on passive surface; (vii) bipolar-unifacial: cores with one surface exploited bidirectionally. Multivariate ordination utilized variables from ref. 15 through *Palaeontological Statistics* software. We normalized variables. The first three components of the PCA explain 80.1% of the variance.

**Reporting summary.** Further information on research design is available in the Nature Research Reporting Summary linked to this article.

## Data availability

All data are available in the main text and the supplementary materials. Materials are to be deposited with the National Museums of Tanzania.

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

## Acknowledgements

First, the authors acknowledge the essential contributions to the knowledge presented herein by the Masai communities at Oldupai Gorge. This work was supported by the Canadian Social Sciences and Humanities Research Council under its Partnership Grant Program no. 895-2016-1017. The Tanzania Commission for Science and Technology authorized this work under permit no. 2018-112-NA-2018-36. The Tanzanian Ministry of Natural Resources and Tourism, through its Antiquities Division, granted us permission to carry out this work (14/2017/2018) and authorities at the Ngorongoro Conservation Area allowed us to enter the protected area (BE.504/620/01/53). The export license for the materials presented in this study were obtained from the Antiquities Division (EA.150/297/01: 5/2018/2019) and the Tanzanian Executive Secretary from the Mining Commission (00001258). Pam Akuku and Palmira Saladié are supported by AGAUR (project no. 2017 SGR-1040) and the URV (2018PFR-URV-B2-91). Pam Akuku's doctoral program is funded by the Canadian Social Sciences and Humanities Research Council. Nina Jablonsky, Bernard Wood, and Michael Lague assisted us with the identification of cercopithecine remains.

## Author contributions

J.M.: conceptualization, methodology, analysis, investigation, resources, data curation, writing original draft, writing review and editing, visualization, supervision, project administration, funding acquisition; P.A.: analysis, investigation; N.B.: conceptualization, writing review and editing, funding acquisition; R.B.: resources, project administration; P.B.: conceptualization, investigation, resources, project administration; A.C.: analysis, investigation; T.C.: analysis, investigation, writing review and editing; S.C.: analysis, investigation, writing review and editing; A.C.T.: conceptualization, methodology, analysis, investigation, writing review and editing, visualization; P.D.: conceptualization, methodology, analysis, investigation, resources, writing review and editing, visualization, funding acquisition; J.F.: conceptualization, analysis, investigation, writing review and editing, visualization, funding acquisition; K.F.: investigation; S.H.: analysis, investigation, writing review and editing; S.H.: conceptualization, methodology, analysis, investigation, resources, writing review and editing, visualization, funding acquisition; J.I.: investigation; M.I.: investigation, resources, project administration; S.K.: investigation, resources; P.L.: investigation, writing review and editing; A.M.: investigation, resources, project administration; A.M.: investigation, resources; L.O.: investigation, resources; R.P.: analysis, investigation, writing review and editing, visualization; P.R.: analysis, investigation, writing review and editing, visualization; S.R.: analysis, investigation, visualization; P.S.: analysis, investigation, visualization; G.S.: writing review and editing, visualization; M.S.: conceptualization, methodology, analysis, investigation, resources, data curation, writing review and editing, visualization, supervision; J.U.: analysis, investigation, writing review and editing, visualization; M.P.: conceptualization, investigation, writing review and editing, visualization, supervision, funding acquisition.

## Funding

## Competing interests

The authors declare no competing interests.
