## [Peer Review File · Nature Communications]

Reviewers' Comments:

Reviewer #1:

Remarks to the Author:

Review of 2020 Oldupai environments and archaeology for Nature Communications

This is an interesting and engaging, data-rich paper on the environmental context of early Oldowan archaeological sites in eastern Africa. The authors make the claim that they have provided the most geographically extensive, chronologically controlled dataset to understand the relationship between human behavior and past environments in eastern Africa ~2 million years ago. I am not a specialist in this particular period, and so cannot precisely evaluate if it is the 'best' record, but it is certainly an impressive one from a key 'type locality' for understanding human evolution: Olduvai/Olduapi Gorge in Tanzania.

The paper is well written, and the data are dense (I mean this as a compliment) and I find their interpretations of the data convincing in terms of reconstructions of age, environment, and past human behaviors. I think that this paper provides a wealth of benchmark (new) data of the highest quality and that it will be well-cited. I do wonder if the authors should trim back their final paragraph a bit. While I appreciate that the data are consistent with hominins occupying variable landscapes, given the spatial scale of movements documented by the authors' raw material source data, we really still don't have a good sense of which environments were really preferred ones and which ones might have been simply 'along the way' to other types of habitats. That is, the occasional use of an environment might be quite different from an early hominins were located most of the time. This doesn't detract too much from the punchline of behavioral adaptability, but I do think that some sort of tempering of the argument would be worthwhile.

My recommendation is 'accept with minor revisions'.

Below are a few minor comments for areas that need a bit of work or clarifications.

Line 71. Why have the authors chosen to term these 'living floors'? This is a VERY contentious term that really requires a LOT of additional data to substantiate. Can some other less loaded term be used? We know that hominins were here and dropped some stone tools, and we know that some animals died in the same area, perhaps as unrelated events. Do we really know anything more from the archaeological data?

Lines 77-78: I don't really understand what the concept of "range of extractions" on lithic artifact. I think that I understand what the authors are getting at (intensity of use), but this term is non-standard and not defined. Is this simply a count of scars on a core? If so, this would be a MINIMUM number of extractions. Or is it from refitting data? Some sort of clarification is needed.

Figure 8: The data axes are unlabeled!

Extended data Figure 2: I don't really understand the point of the PCA comparisons, or at least the implications of the findings. Is it to show that there is a gradual temporal progression in the kinds of stone tools found at Oldowan sites, and that this site pretty much fits in where one would expect it to? Extended data Figure 4: Another PCA plot, and again, I am unclear on what the take-home message is for this figure. Why are some data points connected, and what is the nature of the comparative data set? Is it really only one data point as implied by the figure? If so, this seems insufficient to capture the range of variability out there. This should be an easy fix, but some sort of clarification would be useful.

Reviewer #2:

Remarks to the Author:

Dear authors – thank you for producing such an interesting manuscript which evinces a compelling body of evidence from an interdisciplinary team working on one of the most interesting and historic localities in palaeoanthropology. Despite over a century of investigation, Oldupai Gorge is the gift that keeps on giving, and here your team has done a stellar job of exploiting a range of methods to examine environments in the western part of the gorge, at the interface of Olduvai Bed I and the underlying Ngorongoro Formation.

The evidence presented is for periodic land use across a subset of environments punctuated with times when there is no evidence of hominin activity, but good environmental evidence. This dilutes the central argument somewhat, but what is left is (a) an early Oldowan occurrence at Oldupai which is approximately co-eval with that reported in their reference 37 Habermann et al. 2017 but slightly more extensive (b) 5 additional assemblages over 3 subsequent 'occupation levels' which may postdate the initial one by 100 Ka (c) a lot of really interesting environmental data, some of which corresponds with the artefactual levels, and some of which comes from strata or levels where there is no occupation.

The evidence suggests that the hominins were living in diverse physical and biological environments, with indications that these environments both changed and varied significantly over space and time. The evidence for hominin activity is not spread throughout the sequence, though the environmental evidence is. So the more interesting question is, what is the difference between the environments they WERE inhabiting versus the environments when there is no evidence of hominin presence? The claim that occupation is continuous is not supported – re: the penultimate sentence line 158: The ability to persist through ecological change along a widening range are two behavioral features that enabled hominins to expand beyond Africa. It's not so much persistence that you have demonstrated, as flexibility. The scope of the artefactual assemblages do not support a conclusion of continuous occupation, but of episodic exploitation.

Although you make the argument that environmental/ecological (watch conflation of those terms) information is not thick on the ground prior to Bed I times, you don't cite any of the voluminous work of Bobé, Almseged, Su and others on Omo Shungura Fm, nor (and perhaps more relevant, as it is so much closer to Oldupai) the extensive environmental analyses of the Laetolil and Ndolanya Beds (viz Harrison edited volumes and multiple journal articles). It would also be good to relate your materials to the earlier Lomekwian Industry (Harman et al 2015) in the text. A missed opportunity there as, unlike Bed I and later industries, your artefactual assemblage seems to lack cobbles, as does theirs. This might also be down to temporal differences related to your sites being early – perhaps rounded cobbles were not (yet) available in the Western Gorge, or so early in time. The statement on line 73 – Systems follow non-hierarchical strategies – requires some explanation.

The range of information presented regarding the various methodologies to reconstruct the vegetational environment is clear and supports your conclusions regarding the various habitats. It is an impressive range of techniques which benefit from being examined together, as you have done here. One niggle is that oxygen isotope data are not so straightforward as your statement on lines 115/116 would suggest.

Your abstract requires reorganisation and a better statement of the importance of your conclusions. Various editorial suggestions are written on the scan of the document, which is appended to this review. Specific to your faunal analysis, the faunal list which is in supplementary table 3 should have the taxa in taxonomic rank order (with all the tribes of bovids together), Mammalia separate from reptiles and birds, and the excavation trenches in numerical order. Please check the species names, and use taxonomic rank consistently (e.g. Proboscidea indet, not –ean).

Finally, to address the criteria for publication in this journal,

- The data ARE technically sound
- The results are novel (although they deserve a bit more context re other 'older' sites and the Lomekwian)
- The manuscript is important to scientists in the specific field

My sole reservation pertains to the statement that the paper provides strong evidence for its conclusions; not because there is not strong evidence here but that it doesn't quite support the argument you are making. The recommendation would be to moderate the conclusion and, rather than emphasising continuity and time depth, to focus on the variety and flexibility of hominin ecology which your data show. Your work has the potential to influence thinking in the field not only because of its conclusions but because of the thoroughness of the multi-proxy approach you have used.

Reviewer #3:

Remarks to the Author:

The manuscript explores the environmental context of several Oldupai Oldowan localities. It presents a wealth of multiproxy data from sedimentary, archeological vegetation proxies, and faunal proxies to provide ecological context to archeological and hominin remains. Though there are interesting results and insights, the paper requires serious revision to be considered for publication. In particular, the authors need to carefully revise their language throughout the manuscript so that their statements are clear and leave no ambiguity (see line by line comments). Some of the main figures in the paper are really nice (e.g. fig. 6 and 7) but some seem sloppily/hastily done and lack units, axes, and clear descriptions of what data are in each plot (e.g. 1, 3, 8).

The main concern I have about this paper is that some of the statements are well supported by sedimentary and vegetation proxies (e.g. that hominins at these sites occupied highly disturbed environments after volcanic events). However, there is a lot of data that is presented in this paper that is really poorly described and explored (e.g. faunal enamel data, faunal presence/absence data). It is completely unacceptable to present figures without axes and stratigraphic sections without lithology keys. It is also unlikely that anyone who reads this paper will be an expert in every type of data that is presented. Thus, it is incumbent on the authors to make every figure and table easily understandable (I elaborate about this in the line by line comments below). The methods section is also mixed in terms of its quality with some methods written in fragment sentences and some are light on details. For example, how were the fauna identified? In a museum with references? In the field? Who identified them? Which teeth were identified as satisfactory for stable isotope analysis? Were all teeth used regardless of condition?

I would like to see this paper published because multiproxy explorations of hominin ecology are not terribly common and well worth the effort. For example, the result that hominins occupied disturbance environments—a finding which would not be possible without the combination of archeological, vegetation, and sedimentary data—in and of itself is fascinating. I'd like more discussion of this finding and how it fits into the context of early Pleistocene hominins. How does the technology found at this site compare to others in less disturbed environments, such as Lake Turkana? However, to see this paper published, I would request careful and substantial revision of all of the main figures and text. My line by line comments likely only scratch the surface of the editorial and content work since I am not an expert in all the proxies presented here. I urge the authors to ensure that all of the data are and methods presented in a standardized manner, clearly, and accurately before resubmission.

Line by line comments

TEXT

Lines 4-7 "However, for the earliest phase of human evolution featuring the technology-dependent hominins that shaped our lineage since 2.6 Ma, the Oldowan, there is a dearth of archaeological evidence directly associated with rich chronostratigraphic and environmental datasets amenable to tracking ecological change and adaptation to new physiographic conditions"—sentence needs editing. It is a run on and hard to follow.

Line 5- technology dependent—As far

Line 12- “fresh volcanic landscapes” what does this mean?

Line 18- reference 6 Braun et al., 2010 does not reference “fall back foods” but diverse aquatic resources

Line 19- “The broader behavioural and environmental context of this shift remains elusive, however”

What do you mean by broader in this context? How is looking at a single site broader?

Line 24-25 “Furthermore, correlating outcrop with off-site palaeoecological information is only possible at a gross scale.” This sentence needs more explanation.

Line 35- what is “through phytolith” analysis?

Lines 33-36 How are vegetation proxies more direct indicators than faunal proxies for environment?

Both are organisms within the environment.

Line 95 “Initial hominin colonisation occurred during landscape stabilization.” Please explain how you determine this.

Line 98 “hominins likely found safety from predation, materials for tools, and a fresh water source.”

This statement also needs support. Why would hominins have safety from predation in this landscape?

What fresh water?

Line 101-102 “pioneering Pteridaceae facilitated the reestablishment of woody and grassy communities³⁶.” You mix common and scientific names. It would be beneficial to add the common name to this statement to keep things consistent.

Line 114-116 “Stable isotopes from enamel ($\delta^{13}\text{C}$: -7.2 ‰ to 1.6 ‰) point to herbivores consuming a mixture of C3 and C4 plants, while $\delta^{18}\text{O}$ measurements (-4.7 ‰ to -0.2 ‰) suggest that animals were drinking from a similar water source (Fig. 8, Supplementary Table 2).” I have no idea if this statement is true because there are no axes on this figure

Line 132 “cercopithecines from grassy niches” what is a grassy niche? Do you mean their diet?

Environment? Both? If so, specify dietary or environmental niche

Line 155-156 “Multiproxy data all indicate that early Oldowan hominins pioneered a rapid colonization of geo-settings undergoing drastic changes in hydrological resource distribution and structure, and supporting uneven floras” It is really a subset of your data that really show this, no?

FIGURES AND TABLES

Fig. 1 What do the roman numerals refer to? I think I can figure it out, but it should be clearly explained in the figure caption. Where is the key to lithology?

Fig. 3 What units are the axes in?

Fig. 3 Since the points for lithics are red and the density plot scales to red, it is not possible to see the lithology concentration within the kernel density map. Colors should be revised.

Fig. 7 Really nice!

Fig. 8 Axes are not labelled! This figure needs serious work. The title is “abundance of D13C and D18O” but so far as I can tell the figure is a scatterplot of all individual D13C and D18O values from different taxa. The points are very small and hard to differentiate. Further, the different plots are not clearly labelled. For example, “a-c Values suggest that environmental changes did not markedly impact herbivore feeding habitats >1.9Ma.” Where did these values come from? Why are they in different plots? This figure is not useful without A LOT more information added. D. suggests that you have modern data from somewhere. Where is it? Is this graph entirely modern data? Is there fossil data in it?

Table 1 I am not an expert in mineralogy, but I don't think I should have to be to understand what is going on in this table. Why are there no headers? What are the different numbers? What should I be getting out of this table?

Table 2 is much more clearly explained. Use it as your model.

REVIEWER COMMENTS

We would like to take the opportunity to thank our reviewers for their input.

Reviewer #1 (Remarks to the Author):

Review of 2020 Oldupai environments and archaeology for Nature Communications

This is an interesting and engaging, data-rich paper on the environmental context of early Oldowan archaeological sites in eastern Africa. The authors make the claim that they have provided the most geographically extensive, chronologically controlled dataset to understand the relationship between human behavior and past environments in eastern Africa ~2 million years ago. I am not a specialist in this particular period, and so cannot precisely evaluate if it is the 'best' record, but it is certainly an impressive one from a key 'type locality' for understanding human evolution: Olduvai/Olduapi Gorge in Tanzania. The paper is well written, and the data are dense (I mean this as a compliment) and I find their interpretations of the data convincing in terms of reconstructions of age, environment, and past human behaviors. I think that this paper provides a wealth of benchmark (new) data of the highest quality and that it will be well-cited.

Thank you very much for your support

I do wonder if the authors should trim back their final paragraph a bit. While I appreciate that the data are consistent with hominins occupying variable landscapes, given the spatial scale of movements documented by the authors' raw material source data, we really still don't have a good sense of which environments were really preferred ones and which ones might have been simply 'along the way' to other types of habitats. That is, the occasional use of an environment might be quite different from an early hominins were located most of the time. This doesn't detract too much from the punchline of behavioral adaptability, but I do think that some sort of tempering of the argument would be worthwhile.

We removed all terms that would imply permanence, persistence. In the revised Discussion/Conclusion, we now say:

“The evidence uncovered is of periodic land use across a subset of environments punctuated with times when there is no evidence of hominin activity.”

My recommendation is 'accept with minor revisions'.

Below are a few minor comments for areas that need a bit of work or clarifications.

Line 71. Why have the authors chosen to term these 'living floors'? This is a VERY contentious term that really requires a LOT of additional data to substantiate. Can some other less loaded term be used? We know that hominins were here and dropped some stone tools, and we know that some animals died in the same area, perhaps as unrelated events. Do we really know anything more from the archaeological data?

Thank you for pointing this out. The term 'living floor' has been removed.

Lines 77-78: I don't really understand what the concept of "range of extractions" on lithic artifact. I think that I understand what the authors are getting at (intensity of use), but this term is non-standard and not defined. Is this simply a count of scars on a core? If so, this would be a MINIMUM number of extractions. Or is it from refitting data? Some sort of clarification is needed.

We have detailed this point:

"Intensity of lithic reduction is inferred from the minimum number of extractions per core, ranging from 2-16 removals."

Figure 8: The data axes are unlabeled!

We apologize for this omission during graphic design.

Labels have been included.

Extended data Figure 2: I don't really understand the point of the PCA comparisons, or at least the implications of the findings. Is it to show that there is a gradual temporal progression in the kinds of stone tools found at Oldowan sites, and that this site pretty much fits in where one would expect it to?

We left out inadvertently the PCA loadings and a table that justify the rationale for the PCA. We inserted these two documents as Extended Data Fig. 3 and Supplementary Table 2. As for the main text, rationale and inferences are presented in Line 83:

Principal Component Analysis (PCA, Extended Data Fig. 2a-b), tells that i) Ewass Oldupa's Oldowan compares to that of Kanjera South¹⁴, Fejej¹¹, and Frida Leakey KorongoZinj²¹, and ii) the site has an intermediate position between the oldest Oldowan repertoires >2 Ma and younger sites ≤1.8Ma, where spheroids also abound. Moreover, this PCA, inclusive of 18 assemblages comprising 11 technical variables¹⁵ demonstrates the outlier character of Lomekwi and its lack of affinity with Ewass Oldupa."

Extended data Figure 4: Another PCA plot, and again, I am unclear on what the take-home message is for this figure. Why are some data points connected, and what is the nature of

the comparative data set? Is it really only one data point as implied by the figure? If so, this seems insufficient to capture the range of variability out there. This should be an easy fix, but some sort of clarification would be useful.

We have extended our caption and revised the PCA graph. The comparative dataset comes from the two ecosystems that, as a continuum, and after calibration in past work, would frame the expected ancient vegetation: Zambezian woodland >>>Acacia-Commiphora woodlands. We enhanced the lines of all PCA hulls grouping the phytoliths from different stratigraphic units, and labelled them accordingly. In the caption, we provide additional clarification on what the take home message should be.

Reviewer #2 (Remarks to the Author):

Dear authors – thank you for producing such an interesting manuscript which evinces a compelling body of evidence from an interdisciplinary team working on one of the most interesting and historic localities in palaeoanthropology. Despite over a century of investigation, Oldupai Gorge is the gift that keeps on giving, and here your team has done a stellar job of exploiting a range of methods to examine environments in the western part of the gorge, at the interface of Olduvai Bed I and the underlying Ngorongoro Formation.

Thank you very much for your support

The evidence presented is for periodic land use across a subset of environments punctuated with times when there is no evidence of hominin activity, but good environmental evidence. This dilutes the central argument somewhat, but what is left is (a) an early Oldowan occurrence at Oldupai which is approximately co-eval with that reported in their reference 37 Habermann et al. 2017 but slightly more extensive (b) 5 additional assemblages over 3 subsequent 'occupation levels' which may postdate the initial one by 100 Ka (c) a lot of really interesting environmental data, some of which corresponds with the artifactual levels, and some of which comes from strata or levels where there is no occupation.

The evidence suggests that the hominins were living in diverse physical and biological environments, with indications that these environments both changed and varied significantly over space and time. The evidence for hominin activity is not spread throughout the sequence, though the environmental evidence is. So the more interesting question is, what is the difference between the environments they WERE inhabiting versus the environments when there is no evidence of hominin presence?

We agree this is an important question, but one we cannot answer with the evidence at hand. Thus, we state the question in our revised Conclusion paragraph.

The claim that occupation is continuous is not supported – re: the penultimate sentence line 158: The ability to persist through ecological change along a widening range are two behavioral features that enabled hominins to expand beyond Africa. It's not so much persistence that you have demonstrated, as flexibility.

Agreed, in the Abstract and elsewhere we describe a “record of episodic exploitation” The word ‘continuous’ has been deleted from the manuscript.

The scope of the artefactual assemblages do not support a conclusion of continuous occupation, but of episodic exploitation.

We agree with this, and have changed the wording in our conclusions using terms that do not denote persistence. Instead we refer to: “The ability to occupy variable landscapes and multiple habitats” and “behavioral flexibility”

Although you make the argument that environmental/ecological (watch conflation of those terms) information is not thick on the ground prior to Bed I times, you don't cite any of the voluminous work of Bobé, Almseged, Su and others on Omo Shungura Fm, nor (and perhaps more relevant, as it is so much closer to Oldupai) the extensive environmental analyses of the Laetolil and Ndolanya Beds (viz Harrison edited volumes and multiple journal articles).

We have now cited the seminal work by Bobe et al 2002, and the important volume edited by Harrison et al 2011. We would like to clarify that initially we left them out because the time frame under discussion in our paper, about 2Ma, was a bit removed from the 2.5/2.6 Ma studied in their work. We agree however that these are important references and have added them in Line 52.

It would also be good to relate your materials to the earlier Lomekwian Industry (Harman et al 2015) in the text. A missed opportunity there as, unlike Bed I and later industries, your artefactual assemblage seems to lack cobbles, as does theirs. This might also be down to temporal differences related to your sites being early – perhaps rounded cobbles were not (yet) available in the Western Gorge, or so early in time.

Following the inferences from the PCA Extended Data Fig. 2a-b we show the outlier nature of Lomekwi relative to Ewass; thus,

“Principal Component Analysis (PCA, Extended Data Fig. 2a-b), tells that i) Ewass Oldupa's Oldowan compares to that of Kanjera South¹⁴, Fejej¹¹, and Frida Leakey Korongo Zinj²¹, and ii) the site has an intermediate position between the oldest Oldowan repertoires >2 Ma and younger sites ≤1.8Ma, where spheroids also abound. Moreover, this PCA, inclusive of 18 assemblages

comprising 11 technical variables¹⁵ demonstrates the outlier character of Lomekwi and its lack of affinity with Ewass Oldupa.”

The statement on line 73 – Systems follow non-hierarchical strategies – requires some explanation.

We provide additional clarification in the revised sentence, which reads:

“There is no differential management of knapped surfaces, thus the technological systems do not follow hierarchical reduction strategies.”

The range of information presented regarding the various methodologies to reconstruct the vegetational environment is clear and supports your conclusions regarding the various habitats. It is an impressive range of techniques which benefit from being examined together, as you have done here. One niggle is that oxygen isotope data are not so straightforward as your statement on lines 115/116 would suggest.

We have expanded on the original sentence to reflect the complexity brought up by our referee. Line 150 currently reads:

“Although $\delta^{18}\text{O}$ measurements can be influenced by a number of factors including precipitation, temperature, rainfall source, and physiology, the range below Tuff IA (-4.7 ‰ to -0.2 ‰) is consistent with animals obtaining their water from a similar source and no clear distinctions between taxa are obvious.”

Your abstract requires reorganisation and a better statement of the importance of your conclusions. Various editorial suggestions are written on the scan of the document, which is appended to this review.

The abstract has been thoroughly rewritten. Thank you for your help, and for the annotated .pdf

Specific to your faunal analysis, the faunal list which is in supplementary table 3 should have the taxa in taxonomic rank order (with all the tribes of bovids together), Mammalia separate from reptiles and birds, and the excavation trenches in numerical order. Please check the species names, and use taxonomic rank consistently (e.g. Proboscidea indet, not –ean).

This table has been checked and fixed accordingly.

Finally, to address the criteria for publication in this journal,

- The data ARE technically sound
- The results are novel (although they deserve a bit more context re other ‘older’ sites and the Lomekwian)
- The manuscript is important to scientists in the specific field

My sole reservation pertains to the statement that the paper provides strong evidence for its conclusions; not because there is not strong evidence here but that it doesn't quite support the argument you are making. The recommendation would be to moderate the conclusion and, rather than emphasising continuity and time depth, to focus on the variety and flexibility of hominin ecology which your data show. Your work has the potential to influence thinking in the field not only because of its conclusions but because of the thoroughness of the multi-proxy approach you have used.

Thank you, I hope we have achieved that with our revisions.

Reviewer #3 (Remarks to the Author):

The manuscript explores the environmental context of several Oldupai Oldowan localities. It presents a wealth of multiproxy data from sedimentary, archeological vegetation proxies, and faunal proxies to provide ecological context to archeological and hominin remains. Though there are interesting results and insights, the paper requires serious revision to be considered for publication. In particular, the authors need to carefully revise their language throughout the manuscript so that their statements are clear and leave no ambiguity (see line by line comments).

Thank you for your comments: we have thoroughly edited the manuscript and hope the improvements are clear now.

Some of the main figures in the paper are really nice (e.g. fig. 6 and 7) but some seem sloppily/hastily done and lack units, axes, and clear descriptions of what data are in each plot (e.g. 1, 3, 8).

These figures have been edited: We apologize for this omission during graphic design. Labels have been included.

The main concern I have about this paper is that some of the statements are well supported by sedimentary and vegetation proxies (e.g. that hominins at these sites occupied highly disturbed environments after volcanic events).

However, there is a lot of data that is presented in this paper that is really poorly described and explored (e.g. faunal enamel data, faunal presence/absence data).

We have added details in the Methods section under Stable Carbon and Oxygen isotope analysis to address this concern:

“Specimens selected all had excellent preservation conditions, and no weathered or fragmented dental specimens were considered.”

It is completely unacceptable to present figures without axes and stratigraphic sections without lithology keys.

We have enhanced Fig 1 to show lithologies but also various sedimentary structures

It is also unlikely that anyone who reads this paper will be an expert in every type of data that is presented. Thus, it is incumbent on the authors to make every figure and table easily understandable (I elaborate about this in the line by line comments below). The methods section is also mixed in terms of its quality with some methods written in fragment sentences and some are light on details. For example, how were the fauna identified? In a museum with references? In the field? Who identified them?

We have now provided these details under the Methods section explaining the identification criteria and references. Thank you for pointing this out.

Which teeth were identified as satisfactory for stable isotope analysis? Were all teeth used regardless of condition?

Our response to this comment is provided above this paragraph, on the previous page, where this comment is listed.

I would like to see this paper published because multiproxy explorations of hominin ecology are not terribly common and well worth the effort. For example, the result that hominins occupied disturbance environments—a finding which would not be possible without the combination of archeological, vegetation, and sedimentary data—in and of itself is fascinating. I'd like more discussion of this finding and how it fits into the context of early Pleistocene hominins.

Thank you. We have developed this point further in the discussion / conclusion and echoed this idea in our new title.

How does the technology found at this site compare to others in less disturbed environments, such as Lake Turkana?

We invite the reader to check our revision and the additional data provided: The comparison of our technology with sites across East Africa is elucidated by Principal Component Analysis (PCA, Extended Data Fig. 2a-b), which tells that i) Ewass Oldupa's Oldowan compares to that of Kanjera South¹⁴, Fejej¹¹, and Frida Leakey Korongo Zinj²¹, and ii) the site has an intermediate position between the oldest Oldowan repertoires >2 Ma and younger sites ≤1.8Ma, where spheroids also abound. The Lake Turkana sites are also part of this analysis.

However, to see this paper published, I would request careful and substantial revision of all of the main figures and text. My line by line comments likely only scratch the surface of the editorial and content work since I am not an expert in all the proxies presented here. I urge the authors to ensure that all of the data are and methods presented in a standardized

manner, clearly, and accurately before resubmission. Line by line comments

TEXT

Lines 4-7 “However, for the earliest phase of human evolution featuring the technology-dependent hominins that shaped our lineage since 2.6 Ma, the Oldowan, there is a dearth of archaeological evidence directly associated with rich chronostratigraphic and environmental datasets amenable to tracking ecological change and adaptation to new physiographic conditions”—sentence needs editing. It is a run on and hard to follow.

Addressed

Line 5- technology dependent—As far

Addressed

Line 12- “fresh volcanic landscapes” what does this mean?

To clear this, we now say: “new volcanic landscape”

Line 18- reference 6 Braun et al., 2010 does not reference “fall back foods” but diverse **aquatic resources**

Addressed: We now state: “overall dietary diversification”

Line 19- “The broader behavioural and environmental context of this shift remains elusive, however” What do you mean by broader in this context? How is looking at a single site broader?

We have deleted the term ‘broader’

Line 24-25 “Furthermore, correlating outcrop with off-site palaeoecological information is only possible at a gross scale.” This sentence needs more explanation.

We provide additional information on the use of environmental data from lake cores and boreholes to understand hominin ecology. We currently say:

“Furthermore, extrapolating offsite palaeoecological proxies from penecontemporaneous boreholes and lake-drilling sequences¹⁸ has limited applicability for understanding localized land use, the synchronous / diachronous occupation of varied terrestrial niches, and targeted habitat exploitation by Oldowan hominins.”

Line 35- what is “through phytolith” analysis?

We changed to: “from phytolith analysis”

Lines 33-36 How are vegetation proxies more direct indicators than faunal proxies for environment? Both are organisms within the environment.

We deleted this sentence

Line 95 "Initial hominin colonisation occurred during landscape stabilization." Please explain how you determine this.

In this version, we have explained how this inference was made:

"The Naabi ignimbrite was the result of a high energy, catastrophic volcanic eruption and associated debris flow that drastically reshaped the landscape. After this, the volcanic events are less impactful and stable environments such as river channels and floodplains were able to develop."

Line 98 "hominins likely found safety from predation, materials for tools, and a fresh water source." This statement also needs support. Why would hominins have safety from predation in this landscape? What fresh water?

The revised manuscript reads:

"Initial hominin colonisation, documented in the form of 10 stone tools, appears 1 m above the Naabi ignimbrite and 17 m below Tuff IA. It occurred after landscape stabilisation in association with a sinuous meandering river flowing northwest, as revealed in trench no. 7: The Naabi ignimbrite was the result of a high energy, catastrophic volcanic eruption and associated debris flow that drastically reshaped the landscape. After this, the volcanic events were less impactful and stable environments such as river channels and floodplains were able to develop. In a palaeogeographic setting where the quartzitic basement cropped out from the extruded pyroclastic flow and a fresh water channel ran through the distal part of a volcanoclastic fan, hominins likely diminished the risk from ambush by periodically using the foothill of the inselberg, within short distance of materials for tools as well as the floodplain."

Line 101-102 "pioneering Pteridaceae facilitated the reestablishment of woody and grassy communities³⁶." You mix common and scientific names. It would be beneficial to add the common name to this statement to keep things consistent.

We re-wrote this section:

"The floral context is established by phytoliths (Fig. 5a, 6, Extended Data Fig. 5), in which several fern types³⁵⁻³⁷ dominate assemblages (Fig. 6a,d,e), suggesting the existence of a fern meadow with minor woody growth and grasses; thus, pioneering bracken ferns facilitated the re-establishment of woody and grassy communities after a decrease in destructive volcanic events³⁸

Line 114-116 "Stable isotopes from enamel ($\delta^{13}\text{C}$: -7.2 ‰ to 1.6 ‰) point to herbivores consuming a mixture of C3 and C4 plants, while $\delta^{18}\text{O}$ measurements (-4.7 ‰ to -0.2 ‰)

suggest that animals were drinking from a similar water source (Fig. 8, Supplementary Table 2).” I have no idea if this statement is true because there are no axes on this figure

We apologize for this omission during graphic design.

Labels have been included.

Line 132 “cercopithecines from grassy niches” what is a grassy niche? Do you mean their diet? Environment? Both? If so, specify dietary or environmental niche

In our revised text we make clear that these are:

“Plio-Pleistocene cercopithecines that feed in grassy environments”

Line 155-156 “Multiproxy data all indicate that early Oldowan hominins pioneered a rapid colonization of geo-settings undergoing drastic changes in hydrological resource distribution and structure, and supporting uneven floras” It is really a subset of your data that really show this, no?

Our text now says:

“The earliest Olduvai hominins colonized unstable environments. Sedimentary and vegetation data indicate that early Oldowan hominins pioneered a rapid colonization of geo-settings undergoing drastic changes in hydrological resource distribution and structure, and supporting uneven floras.”

FIGURES AND TABLES

Fig. 1 What do the roman numerals refer to? I think I can figure it out, but it should be clearly explained in the figure caption. Where is the key to lithology?

We have enhanced Fig 1 to show lithologies but also sedimentary structures. Our captions also explains that the column presented under B, had to be broken down in two portions to fit. Roman I is the lower portion, Roman II is the upper portion.

Fig. 3 What units are the axes in?

We have added graphic scales throughout and mentioned the scale is in meters in our caption.

Fig. 3 Since the points for lithics are red and the density plot scales to red, it is not possible to see the lithology concentration within the kernel density map. Colors should be revised.

We changed the color to blue.

Fig. 7 Really nice!

Thanks

Fig. 8 Axes are not labelled! This figure needs serious work. The title is “abundance of D13C and D18O” but so far as I can tell the figure is a scatterplot of all individual D13C and D18O values from different taxa. The points are very small and hard to differentiate. Further, the different plots are not clearly labelled. For example, “a-c Values suggest that environmental changes did not markedly impact herbivore feeding habitats >1.9Ma.” Where did these values come from? Why are they in different plots? This figure is not useful without A LOT more information added. D. suggests that you have modern data from somewhere. Where is it? Is this graph entirely modern data? Is there fossil data in it?

To facilitate reading of the evidence, we have separated the two datasets that were originally put together. One of them, remains as Fig. 8, and it shows the actual stable carbon ($\delta^{13}\text{C}$) and oxygen ($\delta^{18}\text{O}$) measurements of animal teeth from Lower Bed I, below Tuff IA grouped by trenches: T2, T3, and T5.

The other half of former Fig. 8 has become Extended Data Figure 6. This compares stable carbon ($\delta^{13}\text{C}$) and oxygen ($\delta^{18}\text{O}$) measurements of animal teeth from Ewass Oldupa to fossil counterpart data published by others. Note that only data from the same families of taxa available at Lower Bed I have been included in the comparison.

Table 1 I am not an expert in mineralogy, but I don't think I should have to be to understand what is going on in this table. Why are there no headers? What are the different numbers? What should I be getting out of this table? Table 2 is much more clearly explained. Use it as your model.

We apologize for this omission during graphic design. Headers have now been included. Tuff mineral composition fingerprints a specific Tuff, and refers to specific age. In the header of this table, we have added a sentence to help understand the non-specialist in tuff geochemistry that the consistency between our study and previous studies supports the stratigraphic context of Ewass Oldupa.

Reviewers' Comments:

Reviewer #1:

Remarks to the Author:

I have nothing substantial to add here. I think that the authors have done an excellent job addressing my concerns (and those of the other reviewers) in this revised manuscript. It stands as a model of multidisciplinary research, compressing into a single paper what likely would have been a monograph if written 40 years ago.

Reviewer #2:

Remarks to the Author:

Dear authors - thank you for your assiduous attention to many of the comments of your reviewers. The resulting manuscript is an improvement and more consonant with the evidence as presented.

Please reexamine the table of fauna - now Supplementary table 4 - and take on board the suggestions regarding taxonomic lists (there are many published examples of this). If the trenches are not in numerical order for a reason, say why not in the caption. Specimens is misspelled in the heading. The table should be presented in taxonomic order (with all the tribes of bovids together), Mammalia separate from reptiles and birds. Please check the species names (what are H gorgon and P latidens?) and use taxonomic rank consistently (e.g. Proboscidea indet, not -ean). The indets should be at the bottom, not in alphabetical order.

Other niggles centre on the use of terms for land and habitat surface use which imply duration / intensity, e.g. colonise, including in the title. Use/Exploit probably better, especially for the older occurrences.

Here are some other minor suggestions:

Line 28 - consider 'dietary innovations' for 'new diets'

Line 32 - lacuna; lacunae is plural - does it fill it? Partially? address?

Line 33 fast > rapidly

Line 36 throughout > during

Line 39 colonisation not the right word; suggest changing line to 'suggesting hominins used emerging volcanic landscapes and disturbance biomes . . .'

Line 50 niche > environments or habitats

Line 87 - insert 'DK I' somewhere as this is what it's commonly known as.

Line 105 - likewise 'FLK Zinj or 22'

Line 119 - occupation > use? Presence? It's 10 artifacts!

Line 122 - colonisation > use? Presence?

Line 136 - does this date have no error?

Line 161 - you've deleted the word 'permanent'; perhaps something saying why you think there is a water source? Sedimentology? Presence of water-dependent taxa? Seasonally/ permanently available water source?

Line 164 - places > sources

Line 203 - colonised again - used? Occupied?

Line 205 - colonized..... exploited?

Reviewer #3:

Remarks to the Author:

I'd like to first reiterate that this is a wonderful multiproxy dataset that, presented together, gives a

novel perspective about early hominin ecology. Overall the manuscript and the figures are improved but both still lack (in some cases) a careful/explicit discussion of the ecological implications of the multiproxy results. For example, here is a very long discussion of the geological information in the results but only one sentence each about the leaf wax biomarker and enamel stable isotope results that convey very little information. How does what you found for these proxies compare to other studies of this region or time period? If you are space limited, consider making your geological descriptions more succinct and referring to the extended data. The abstract is more direct but needs editing for clarity. The abstract would also benefit from more explicit explanation of the main results that affect your assessment of hominin behavior from your various ecological proxies.

My line by line comments offer some suggestions for the abstract, results and discussion sections but, as I suggested previously, there should be a thoughtful discussion of all the different data sources presented in this manuscript. If a proxy does not provide useful information, perhaps it should not be included. Further, overall the figure captions still need more detail. Every aspect of the figures that is important for understanding the results MUST be mentioned. I've made suggestions about the main figures in the line by line comments, but the extended data figures would also benefit from this attention to detail.

Line by line comments (line numbers are from the non-tracked changes version of the new manuscript):

Line 28: Rapid environmental change is a known catalyst for human evolution; also driving new diets, habitat diversification, and dispersal.

Line 29: As far as technology-dependent hominins are concerned, There is a dearth of information between 2.6 - 1.9 Ma amenable for assessing Oldowan adaptations to new physiographic conditions. Note: What is an "Oldowan adaptation?" Do you mean adaptations to the Oldowan?

Line 33-35: The earliest Oldupai hominins inhabited floodplains of small sinuous channels, and later river-influenced contexts that included the oldest palaeolake setting documented in outcrop.

Note: Which out crop? Those associated with hominins? All of the outcrops at Oldupai? If all, you should say "documented at the site."

Line 57: Add the word dataset and clarify what you mean by "Oldowan behavior". Suggested modified sentence: This dataset clarifies the environmental context of Oldowan-bearing hominin behavior by examining the associated faunal communities, stable isotopes from enamel, and also provides...

Line 89: "1373 fossils" Is this the number of specimens? The number of individuals? Please clarify.

Line 98: "number of cores 19"—I think this should be "number of cores = 19"

103: Though you describe your PCA in the extended figures and tables, the text describing the PCA in the main text is too sparse. You need to say what about the PCA suggests that Ewass Oldupa is similar to the other sites you mention. Saying that it "compares" to the other sites does not convey any information about what compares.

130: "hominins likely diminished the risk from ambush by periodically using the foothill of the inselberg" This is still too speculative. They may have diminished the risk of ambush but there is no evidence that is presented to support this so the statement should be qualified.

147: "Long chain (C29 C31 C33) terrestrial plant biomarkers from leaf wax exhibit an environmental context of open woodlands (Fig. 7a-c)." From where? Is this the average signal from the whole assemblage?

148: "Stable isotopes from enamel ($\delta^{13}C$: -7.2‰ to 1.6 ‰) point to herbivores consuming a mixture of C3 and C4 plants (Fig. 8)." Same comment as above. Are these enamel isotopes from the whole

assemblage or from particular trenches? How does this line up with the other vegetation data sources?

187: "Ewass Oldupa is a high resolution, multi-episode, stratified site that precedes and straddles Bed I, indicating occupation of broad-spectrum habitats across multiple ecological niches." This sentence does not convey much information to me. Whose ecological niches do you refer to? What is a broad-spectrum habitat? You have a really interesting dataset. Refer to more of the details, for example, that hominins or their cultural remains are associated habitats from post volcanic disturbance habitats to hyper-arid habitats.

193: "Oldowan ecology associates with meandering rivers" What is Oldowan ecology?

Line 199-200: "The Oldowan toolkits from this time intersect earlier and contemporaneous technologies from Ethiopia and Kenya, while advancing tool types such as the spheroid commonly seen in younger assemblages²⁴." This sentence does not make sense to me. Perhaps it should be two sentences?

Lines 208-210: "Our work reveals early Oldowan hominins that lived in diverse physical and biological environments, with indications that these environments both changed and varied significantly over space and time." New evidence of hominin remains is not presented here, which is what this sentence suggests. Also changed and varied is repetitive.

Lines 211-213: "The ability to occupy variable landscapes along a widening range are two behavioral features that enabled hominins to expand beyond Africa." What do you mean by widening range?

Figures

Figure 1: "Location map." It would be helpful to state what is starred in the caption even though it is mentioned in the map and why you have included the other localities (something as simple as, "associated lithic sites"—are they all of the same time period?). "Stratigraphic section" Much improved! But is this a composite? Also, it would add clarity to this caption if you added the associated roman numeral when you mention the different beds. For example, rather than, "Lower bed I—Six units...", state, "(i) Lower bed I—Six units..."

Figure 2: It would be useful to note which beds these features are associated with.

Figure 7: What are the Roman numerals in part A? "B-F The relative abundance of each n-alkane compound, their $\delta^{13}\text{C}$ values..." Same question as in the main text. Where do these samples come from? Are B-F averaged values from all soils from both the lower and upper sections?

Extended Fig. 5-D: Y-axis is not labelled

REVIEWER COMMENTS

Response

We would like to thank the Reviewers for their immensely encouraging and helpful comments in relation to our manuscript. It was a pleasure to read three Reviews that made highly constructive comments that, we believe, have improved our paper a lot. We have addressed each and every one of the comments and suggestions raised as can be seen in the detailed point by point list below.

Reviewer #1 (Remarks to the Author):

I have nothing substantial to add here. I think that the authors have done an excellent job addressing my concerns (and those of the other reviewers) in this revised manuscript. It stands as a model of multidisciplinary research, compressing into a single paper what likely would have been a monograph if written 40 years ago.

Response

Thank you so much. It makes a difference having your support.

Reviewer #2 (Remarks to the Author):

Dear authors - thank you for your assiduous attention to many of the comments of your reviewers. The resulting manuscript is an improvement and more consonant with the evidence as presented.

Response

Thank you very much

Please reexamine the table of fauna - now Supplementary table 4 - and take on board the suggestions regarding taxonomic lists (there are many published examples of this). If the trenches are not in numerical order for a reason, say why not in the caption.

Response

Our trenches are now in numerical order.

Specimens is misspelled in the heading.

Response

This typo has been fixed.

The table should be presented in taxonomic order (with all the tribes of bovids together)

Response

The table is now in taxonomic order, and all bovids together.

Mammalia separate from reptiles and birds.

Response

Done

*Please check the species names (what are *H gorgon* and *P latidens*?)*

Response

These two typos have been fixed, our apologies.

*and use taxonomic rank consistently (e.g. *Proboscidea indet*, not *-ean*).*

Response

Done

The indets should be at the bottom, not in alphabetical order.

Response

Done

Other niggles centre on the use of terms for land and habitat surface use which imply duration / intensity, e.g. colonise, including in the title. Use/Exploit probably better, especially for the older occurrences. Here are some other minor suggestions:

Line 28 – consider ‘dietary innovations’ for ‘new diets’

Line 32 – lacuna; lacunae is plural – does it fill it? Partially? address?

Line 33 fast > rapidly

Line 36 throughout > during

Line 39 colonisation not the right word; suggest changing line to ‘suggesting hominins used emerging volcanic landscapes and disturbance biomes . . .’

Line 50 niche > environments or habitats

Line 87 – insert ‘DK I’ somewhere as this is what it’s commonly known as.

Line 105 – likewise ‘FLK Zinj or 22’

Line 119 – occupation > use? Presence? It’s 10 artifacts!

Line 122 – colonisation > use? Presence?

Line 136 – does this date have no error?

Line 161 – you’ve deleted the word ‘permanent’; perhaps something saying why you think there is a water source? Sedimentology? Presence of water-dependent taxa? Seasonally/ permanently available water source?

Line 164 – places > sources

Line 203 – colonised again – used? Occupied?

Line 205 – colonized..... exploited?

Response

All these are cases of grammatical correction, word choice, or missing labels, and every single correction has been made. Thank you.

Reviewer #3 (Remarks to the Author):

I’d like to first reiterate that this is a wonderful multiproxy dataset that, presented together, gives a novel perspective about early hominin ecology.

Response

Thank you very much

Overall the manuscript and the figures are improved but both still lack (in some cases) a careful/explicit discussion of the ecological implications of the multiproxy results. For example, here is a very long discussion of the geological information in the results but only one sentence each about the leaf wax biomarker and enamel stable isotope results that convey very little information. How does what you found for these proxies compare to other studies of this region or time period?

Response

We have expanded our discussion of leaf waxes and enamel isotopes in text. These have greatly improved the manuscript, and we thank the reviewer for his/her attention to detail and a need for a comprehensive presentation of the manuscript. The changes we made are in Lines 152-173.

The abstract is more direct but needs editing for clarity. The abstract would also benefit from more explicit explanation of the main results that affect your assessment of hominin behavior from your various ecological proxies.

Response

We edited the abstract for clarity in semantics. Thank you for making us bring out the assessment of hominin ecology based on our results. Much improved.

My line by line comments offer some suggestions for the abstract, results and discussion sections but, as I suggested previously, there should be a thoughtful discussion of all the different data sources presented in this manuscript.

Further, overall the figure captions still need more detail. Every aspect of the figures that is important for understanding the results MUST be mentioned. I've made suggestions about the main figures in the line by line comments, but the extended data figures would also benefit from this attention to detail.

Response

All figure captions have been carefully examined and revised as per instructions. We believe the revisions improved the detail that was presented in the captions in order to provide all necessary information for full comprehension of the figure. See details in the figure caption sections down below

Line by line comments (line numbers are from the non-tracked changes version of the new manuscript):

Line 28: Rapid environmental change is a known catalyst for human evolution; also driving new diets, habitat diversification, and dispersal.

Response

This has been corrected.

Line 29: As far as technology-dependent hominins are concerned, There is a dearth of information between 2.6 - 1.9 Ma amenable for assessing Oldowan adaptations to new physiographic conditions. Note: What is an "Oldowan adaptation?" Do you mean adaptations to the Oldowan?

Response

We have rephrased our sentence to improve clarity.

Line 33-35: The earliest Oldupai hominins inhabited floodplains of small sinuous channels, and later river-influenced contexts that included the oldest palaeolake setting documented in outcrop. Note: Which out crop? Those associated with hominins? All of the outcrops at Oldupai? If all, you should say "documented at the site."

Response

This has been corrected.

Line 57: Add the word dataset and clarify what you mean by “Oldowan behavior”. Suggested modified sentence: This dataset clarifies the environmental context of Oldowan-bearing hominin behavior by examining the associated faunal communities, stable isotopes from enamel, and also provides...

Response

We have rephrased our sentence to improve clarity.

Line 89: “1373 fossils” Is this the number of specimens? The number of individuals? Please clarify.

Response

Done

Line 98: “number of cores 19”—I think this should be “number of cores = 19”

Response

Done

103: Though you describe your PCA in the extended figures and tables, the text describing the PCA in the main text is too sparse. You need to say what about the PCA suggests that Ewass Oldupa is similar to the other sites you mention. Saying that it “compares” to the other sites does not convey any information about what compares.

Response

We have rephrased our sentence to improve clarity, and added the specific characteristics that relate assemblages to one another. See Line 107 and following.

130: “hominins likely diminished the risk from ambush by periodically using the foothill of the inselberg” This is still too speculative. They may have diminished the risk of ambush but there is no evidence that is presented to support this so the statement should be qualified.

Response

We deleted this speculative statement. Thank you.

147: “Long chain (C29 C31 C33) terrestrial plant biomarkers from leaf wax exhibit an environmental context of open woodlands (Fig. 7a-c).” From where? Is this the average signal from the whole assemblage?

Response

In Line 154 we now detail the nature of the signal, which is from weighted averages for each individual sample.

148: “Stable isotopes from enamel ($\delta^{13}C$: -7.2‰ to 1.6 ‰) point to herbivores consuming a mixture of C3 and C4 plants (Fig. 8).” Same comment as above. Are these enamel isotopes from the whole assemblage or from particular trenches? How does this line up with the other vegetation data sources?

Response

We now detail the nature of the signal and provenance in Lines 161 and following, and also provide a comparison with contemporaneous datasets and other vegetation proxies in the same paragraph, along with Fig. 7, 8, and Extended Data Fig. 6.

187: “Ewass Oldupa is a high resolution, multi-episode, stratified site that precedes and straddles Bed I, indicating occupation of broad-spectrum habitats across multiple ecological niches.” This sentence does not convey much information to me. Whose ecological niches do you refer to? What is a broad-spectrum habitat?

Response

We simplified this sentence to clarify meaning.

You have a really interesting dataset. Refer to more of the details, for example, that hominins or their cultural remains are associated habitats from post volcanic disturbance habitats to hyper-arid habitats.

Response

Thank you, we have made the reviewer’s point clear in our new Abstract.

193: “Oldowan ecology associates with meandering rivers” What is Oldowan ecology?

Response

We have rephrased our sentence to improve clarity and say that humans provisioned near meandering rivers instead. Line 210.

Lines 208-210: “Our work reveals early Oldowan hominins that lived in diverse physical and biological environments, with indications that these environments both changed and varied significantly over space and time.” New evidence of hominin remains is not presented here, which is what this sentence suggests. Also changed and varied is repetitive.

Response

We have rephrased our sentence to improve clarity.

Lines 211-213: “The ability to occupy variable landscapes along a widening range are two behavioral features that enabled hominins to expand beyond Africa.” What do you mean by widening range?

Response

We have rephrased our sentence to improve clarity.

Figures

All figure captions have been carefully examined and revised as per instructions. We believe the revisions improved the detail that was presented in the captions in order to provide all necessary information for full comprehension of the figure.

Figure 1: “Location map.” It would be helpful to state what is starred in the caption

Response

We have added this information.

even though it is mentioned in the map and why you have included the other localities (something as simple as, “associated lithic sites”—are they all of the same time period?).

Response

We have added this information.

“Stratigraphic section” Much improved! But is this a composite?

Response

We have added the term ‘single stratigraphic section’ to clarify that this is not a composite.

Also, it would add clarity to this caption if you added the associated roman numeral when you mention the different beds. For example, rather than, “Lower bed I—Six units...,” state, “(i) Lower bed I—Six units...”

Response

We have rephrased our sentence to improve clarity.

Figure 2: It would be useful to note which beds these features are associated with.

Response

Done.

Figure 7: What are the Roman numerals in part A? “B-F The relative abundance of each n-alkane compound, their $\delta^{13}C$ values...” Same question as in the main text. Where do these samples come from? Are B-F averaged values from all soils from both the lower and upper sections?

Response

The caption to Fig. 7 is now expanded and has addressed the question from the reviewer. Thank you.

Extended Fig. 5-D: Y-axis is not labelled

Response

Done.

Thank you

Sincerely,

The authors